# Time series recovery from partial observations via Nonnegative Matrix Factorization

## Abstract

In the modern analysis of time series, one may want to forecast several hundred or thousands of times series with possible missing entries. We introduce a novel algorithm to address these issues, referred to as Sliding Mask Method (SMM). SMM is a method based on the framework of predicting a time window and using completion of nonnegative matrices. This new procedure combines nonnegative factorization and matrix completion with hidden values (*i.e.*, a partially observed matrix). From a theoretical point of view, we prove the statistical guarantees on uniqueness and robustness of the solutions of the completion of partially observed nonnegative matrices. From a numerical point of view, we present experiments on real-world and synthetic data-set confirm forecasting accuracy for the novel methodology.

## 1 Introduction

This article investigates the forecasting of several times series from partial observations. We study times series for which one can provide a lower bound on the observations. In this case, one can assume that the times series are entry-wise *nonnegative*, and one can exploit Nonnegative Matrix Factorization (NMF) approaches, see for instance Paatero & Tapper (1994) and Lee & Seung (1999). For further details, we refer the interested reader to the surveys Wang & Zhang (2013); Gillis (2015; 2017) and references therein. NMF has been widely used in the contexts of document analysis Xu et al. (2003); Essid & Fevotte (2013), hidden Markov chain Fu et al. (1999), representation learning in image recognition Lee & Seung (1999), community discovery Wang et al. (2011), and clustering methods Turkmen (2015). This paper introduces a novel NMF-like procedure for forecasting of several time series. Forecasting for temporal time series has been previously done before through a mixed linear regression and matrix factorization as in Yu et al. (2016), matrix completion for one temporal time series as in Gillard & Usevich (2018), and tensor factorization as in de Araujo et al. (2017); Yokota et al. (2018); Tan et al. (2016).

Our proposed method, the Sliding Mask Method (SMM), inputs the forecast values and it can be viewed as a nonnegative matrix completion algorithm under low nonnegative rank assumption. This framework raises two issues. A first question is the uniqueness of the decomposition, also referred to as *identifiability* of the model. In Theorem 7, we introduce a new condition that ensures uniqueness from partial observation of the target matrix **M**. An other challenge, as pointed by Vavasis (2009) for instance, is that solving *exactly* NMF decomposition problem is NP-hard. Nevertheless NMF-type problems can be solved efficiently using (accelerated) proximal gradient descent method Parikh & Boyd (2013) for block-matrix coordinate descent in an *alternating projection scheme*, *e.g.*, Javadi & Montanari (2020a) and references therein. We rely on these techniques to introduce algorithms inputting the forecast values based on NMF decomposition, see Section 2.3. Theorem 10 complements the theoretical analysis by proving the robustness of the solutions of NMF-type algorithms when entries are missing or corrupted by noise.

**Notation:** Denote by $A^\top$ the transpose of matrix $A$. We use $\mathbb{R}_+^{n \times p}$ to denote $n \times p$ nonnegative matrices. It would be useful to consider the columns description $A_k \in \mathbb{R}^{n_1}$ of matrix $\mathbf{A} = [A_1 \cdots A_{n_2}]$ and the row decomposition $A^{(k)} \in \mathbb{R}^{n_2}$ of a matrix $\mathbf{A}$ using $\mathbf{A}^\top = [(A^{(1)})^\top \cdots (A^{(n_1)})^\top]$ for $\mathbf{A} \in \mathbb{R}^{n_1 \times n_2}$. Notation $A_{i,j}$ indicates the elements of matrix $\mathbf{A}$; $[n]$ represents the set $\{1, 2, \ldots, n\}$; $\mathbf{1}_d$ is the all-ones vector of size $d$; and $\mathbb{1}_{\mathcal{A}}$ is the indicator function of $\mathcal{A}$, such that $\mathbb{1}_{\mathcal{A}} = 0$ if condition $\mathcal{A}$ is verified, $\infty$ otherwise.

## 1.1 The problem of forecasting several time series

This article considers $N \geq 1$ times series on the same temporal period of length $T \geq 1$ in a setting where $N$ and $T$ could be such that $N \geq T$ and possibly $N \gg T$. We would like to forecast the next $F \geq 1$ times. Additionally, one may also aim at reduce the ambient dimension $N \times T$ while maintaining a good approximation of these times series. The observed times series can be represented as a matrix $\mathbf{M}$ of size $N \times T$. A row $\mathbf{M}^{(i)}$ of $\mathbf{M}$ represents a time series and a column $\mathbf{M}_j$ of $\mathbf{M}$ represents timestamp records. We assume that there exists a *target* matrix $\mathbf{M}^\star$ with

$$\mathbf{M} = \mathbf{M}_T^\star + \mathbf{E},$$

where $\mathbf{E}$ is some noise term and $\mathbf{M}_T^\star \in \mathbb{R}_+^{N \times T}$ is a block given by

$$\mathbf{M}^\star := \Big[ \underbrace{\mathbf{M}_T^\star}_{\text{past}} \ \underbrace{\mathbf{M}_F^\star}_{\text{future}} \Big]$$

of the *target matrix* of size $N \times (T + F)$. We decompose the target matrix using the timestamps up to time $T$, namely $\mathbf{M}_T^\star \in \mathbb{R}_+^{N \times T}$, and the timestamps of the future time period $\mathbf{M}_F^\star \in \mathbb{R}_+^{N \times F}$ to be forecasted. The statistical task is the following: given the observation $\mathbf{M}$ predict the future target values $\mathbf{M}_F^\star$, and incidentally a low dimensional representation $\mathbf{M}_T^\star$.

## 1.2 The nonnegative analysis and the archetypal analysis

We aim to decompose a nonnegative matrix $\mathbf{M} \in \mathbb{R}_+^{N \times T}$ as the product of nonnegative matrix $\mathbf{W} \in \mathbb{R}^{N \times K}$ and matrix $\mathbf{H} \in \mathbb{R}^{K \times T}$ by minimizing the Frobenius norm of the difference between the matrix $\mathbf{M}$ and the reconstructed matrix $\widehat{\mathbf{M}} := \mathbf{WH}$, as in Cichocki et al. (2009),

$$\min_{\mathbf{W} \geq \mathbf{0}, \mathbf{H} \geq \mathbf{0}} \|\mathbf{M} - \mathbf{WH}\|_F^2, \tag{NMF}$$

which is convex in $\mathbf{W}$ (with fixed $\mathbf{H}$) and $\mathbf{H}$ (with fixed $\mathbf{W}$), but not convex in both varaibles. Another approach consists in the *archetypal analysis*:

$$\min_{\substack{\mathbf{W} \geq \mathbf{0}, \mathbf{W1}=\mathbf{1} \\ \mathbf{V} \geq \mathbf{0}, \mathbf{V1}=\mathbf{1}}} \|\mathbf{M} - \mathbf{WH}\|_F^2 + \lambda \|\mathbf{H} - \mathbf{VM}\|_F^2, \tag{AMF}$$

where $\lambda > 0$ is a tuning parameter, see for instance Javadi & Montanari (2020a). Different normalisation and constraints can be considered, we exhibit seven variants (see Table 7 in Appendix). We will be particularly interested in

$$\min_{\substack{\mathbf{W} \geq \mathbf{0}, \mathbf{W1}_K=\mathbf{1}_N \\ \mathbf{H} \geq \mathbf{0}}} \|\mathbf{M} - \mathbf{WH}\|_F^2. \tag{NNMF}$$

In (NMF), the sample given by the rows of $\mathbf{M}$ are represented by a conic combination (*i.e.,* sum of nonnegative weights) of the rows of $\mathbf{H}$. Archetypal analysis (AMF) penalises the reciprocal: the rows of $\mathbf{H}$ should be represented as a convex combination of the sample points given by the rows of $\mathbf{M}$. Note that, both parameters $K$ and $\lambda$ can be tuned by cross-validation Arlot & Celisse (2010), as done in our experiments, see Section 3.

## 1.3 Contribution: Nonnegative matrix completion for the analysis of several time series

Given the observation of several time series $\mathbf{M}$ such that $\mathbf{M} = \mathbf{M}_T^\star + \mathbf{E}$, one would like to estimate the target forecast $\mathbf{M}_F^\star$ where $\mathbf{M}^\star := [\mathbf{M}_T^\star \ \mathbf{M}_F^\star]$. We will use a nonnegative matrix factorization to address it. This factorization can be obtained by solving a block convex program, updating $\mathbf{W}$ then $\mathbf{H}$ and repeating the step. This class of algorithms can handle linear constraints. We will therefore allow ourselves to perform a linear transformation on $\mathbf{M}$ to predict $\mathbf{M}_F^\star$. In particular, the operation which consists in taking a matrix $\mathbf{V}$ and looking at the successive values given by a sliding window running through $\mathbf{V}$ in the lexicographic order, is a linear operation.

**General framework:**  We consider any injective linear map $\boldsymbol{\Phi} : \mathbb{R}^{N \times (T+F)} \to \mathbb{R}^{n \times p}$ with $n \geq N$ and $p \geq F$ such that

$$\forall \mathbf{A} \in \mathbb{R}^{N \times T}, \mathbf{B} \in \mathbb{R}^{N \times F}, \quad \boldsymbol{\Phi}([\mathbf{A}\ \mathbf{B}]) = \left[ \begin{array}{c|c} \boldsymbol{\Phi}_1(\mathbf{A}) & \boldsymbol{\Phi}_2(\mathbf{A}) \\ \hline \boldsymbol{\Phi}_3(\mathbf{A}) & \mathbf{B} \end{array} \right],$$

where $\boldsymbol{\Phi}_k$ $(k = 1, 2, 3)$ are linear maps.

**Remark 1** *An example of a linear map relevant for the forecast of several time series is presented in Section 2, with $n := (B - W + 1)N > N$ and $p := WP > F$.*

We denote by

$$\mathbf{X}_0 = \boldsymbol{\Phi}(\mathbf{M}^\star), \ \mathbf{X} = \boldsymbol{\Phi}([\mathbf{M}\ \mathbf{0}_{N \times F}]),$$
$$\mathbf{X}^\star = \boldsymbol{\Phi}([\mathbf{M}_T^\star\ \mathbf{0}_{N \times F}]) \quad \text{and} \quad \mathbf{F} = \boldsymbol{\Phi}([\mathbf{E}\ \mathbf{0}_{N \times F}]).$$

Note that the identity $\mathbf{M} = \mathbf{M}_T^\star + \mathbf{E}$ implies that $\mathbf{X} = \mathbf{X}^\star + \mathbf{F}$. Consider a so-called *mask* operator $\mathbf{T}(\mathbf{N})$ that sets to zero $N \times F$ values of an $n \times p$ matrix $\mathbf{N}$. Namely, given $\mathbf{N} \in \mathbb{R}^{n \times p}$, we define

$$\mathbf{T}(\mathbf{N}) = \left[ \begin{array}{c|c} \mathbf{N}_1 & \mathbf{N}_2 \\ \hline \mathbf{N}_3 & \mathbf{0}_{N \times F} \end{array} \right] \quad \text{and} \quad \mathbf{T}^\perp(\mathbf{N}) = \left[ \begin{array}{c|c} \mathbf{0} & \mathbf{0} \\ \hline \mathbf{0} & \mathbf{N}_4 \end{array} \right],$$

where $(\mathbf{N}_i)_{i=1}^4$ are blocks of $\mathbf{N} = \left[ \begin{array}{c|c} \mathbf{N}_1 & \mathbf{N}_2 \\ \hline \mathbf{N}_3 & \mathbf{N}_4 \end{array} \right]$. Note that $\mathbf{T}(\mathbf{X}_0) = \mathbf{X}^\star$.

**A nonnegative matrix completion problem:**  Our goal is the following matrix completion problem: *Given a noisy and incomplete observation*

$$\mathbf{X} = \mathbf{T}(\mathbf{X}_0) + \mathbf{F}, \tag{1}$$

*where $\mathbf{F}$ is some noise term, find a good estimate of the target $\mathbf{X}_0$.*

We introduce the Mask NNMF:

$$\min_{\substack{\mathbf{W}\mathbf{1}=\mathbf{1}, \mathbf{W} \geq \mathbf{0} \\ \mathbf{H} \geq \mathbf{0} \\ \mathbf{T}(\mathbf{N})=\mathbf{X}}} \|\mathbf{N} - \mathbf{W}\mathbf{H}\|_F^2, \tag{mNMF}$$

where solutions $\mathbf{N} \in \mathbb{R}^{n \times p}$ are such that $\mathbf{T}(\mathbf{N}) = \mathbf{X}$ (observed values) and $\mathbf{T}^\perp(\mathbf{N}) = \mathbf{T}^\perp(\mathbf{W}\mathbf{H})$ (forecast values). This latter formulation is an instance of Matrix Completion Nguyen et al. (2019). Forecasting problem reduces to Matrix Completion problem, whose aim is finding the nonnegative matrix factorization $\mathbf{N} \simeq \mathbf{W}\mathbf{H}$ of observed matrix $\mathbf{X}$ such that $\mathbf{T}(\mathbf{N}) = \mathbf{X}$.

**Remark 2** *Problem* (mNMF) *is NNMF when* $\mathbf{T} = \mathbb{I}$*, where $\mathbb{I}$ is the identity operator.*

One can drop the constraint $\mathbf{H} \geq 0$ which leads to an other approach referred to as Mask AMF:

$$\min_{\substack{\mathbf{W} \geq \mathbf{0}, \mathbf{W}\mathbf{1}=\mathbf{1} \\ \mathbf{V} \geq \mathbf{0}, \mathbf{V}\mathbf{1}=\mathbf{1} \\ \mathbf{T}(\mathbf{N})=\mathbf{X}}} \|\mathbf{N} - \mathbf{W}\mathbf{H}\|_F^2 + \lambda \|\mathbf{H} - \mathbf{V}\mathbf{N}\|_F^2 \tag{mAMF}$$

**Remark 3** *When* $\mathbf{T} = \mathbb{I}$*, Problem* (mAMF) *reduces to standard AMF formulation (AMF).*

### 1.3.1 Uniqueness from partial observations

Recall that $\mathbf{X}^\star$ is the *mask* of $\mathbf{X}_0$ since

$$\mathbf{X}^\star = \mathbf{T}(\mathbf{X}_0) = \left[ \begin{array}{c|c} \mathbf{X}_1 & \mathbf{X}_2 \\ \hline \mathbf{X}_3 & \mathbf{0}_{N \times F} \end{array} \right],$$

where $\mathbf{X}_1 \in \mathbb{R}^{(n-N) \times (p-F)}$, $\mathbf{X}_2 \in \mathbb{R}^{(n-N) \times F}$, and $\mathbf{X}_3 \in \mathbb{R}^{N \times (p-F)}$ are blocks of $\mathbf{X}_0$. Let us consider

$$\mathbf{T}_{\text{train}}(\mathbf{X}_0) := [\mathbf{X}_1\ \mathbf{X}_2], \quad \mathbf{T}_{\text{test}}(\mathbf{X}_0) := [\mathbf{X}_3\ \mathbf{0}_{N \times F}],$$
$$\mathbf{T}_T(\mathbf{X}_0) := \left[ \begin{array}{c} \mathbf{X}_1 \\ \mathbf{X}_3 \end{array} \right], \qquad \mathbf{T}_F(\mathbf{X}_0) := \left[ \begin{array}{c} \mathbf{X}_2 \\ \mathbf{0}_{N \times F} \end{array} \right].$$

**Remark 4** *Let* $\mathbf{X}_0 := \mathbf{W}_0 \mathbf{H}_0$, $\mathbf{H}_0 := [\mathbf{H}_{0T} \ \mathbf{H}_{0F}]$, *and* $\mathbf{W}_0^\top := [\mathbf{W}_{0\text{train}}^\top \ \mathbf{W}_{0\text{test}}^\top]$, *then*

$$\mathbf{T}_{\text{train}}(\mathbf{X}_0) = \mathbf{W}_{0\text{train}} \mathbf{H}_0 \,, \quad \mathbf{X}_3 = \mathbf{W}_{0\text{test}} \mathbf{H}_{0T} \,,$$
$$\mathbf{T}_T(\mathbf{X}_0) = \mathbf{W}_0 \mathbf{H}_{0T} \,, \quad\quad \mathbf{X}_2 = \mathbf{W}_{0\text{train}} \mathbf{H}_{0F} \,.$$

A first issue is the *uniqueness* of the decomposition $\mathbf{W}_0 \mathbf{H}_0$ *given partial observations*, namely proving that *Partial Observation Uniqueness* (**POU**) property holds:

$$\text{If } \mathbf{T}(\mathbf{W}\mathbf{H}) = \mathbf{T}(\mathbf{W}_0 \mathbf{H}_0) \text{ Then } (\mathbf{W}, \mathbf{H}) \equiv (\mathbf{W}_0, \mathbf{H}_0) \,, \quad\quad \textbf{(POU)}$$

where $\equiv$ means up to positive scaling and permutation: if an entry-wise nonnegative pair $(\mathbf{W}, \mathbf{H})$ is given then $(\mathbf{W}\mathbf{P}\mathbf{D}, \mathbf{D}^{-1}\mathbf{P}^\top \mathbf{H})$ is also a nonnegative decomposition $\mathbf{W}\mathbf{H} = \mathbf{W}\mathbf{P}\mathbf{D} \times \mathbf{D}^{-1}\mathbf{P}^\top \mathbf{H}$, where $\mathbf{D}$ scales and $\mathbf{P}$ permutes the columns (resp. rows) of $\mathbf{W}$ (resp. $\mathbf{H}$). When we observe the full matrix $\mathbf{X}_0 = \mathbf{W}_0 \mathbf{H}_0$, the issue on uniqueness has been addressed under some sufficient conditions on $\mathbf{W}, \mathbf{H}$, *e.g.*, *Strongly boundary closeness* of Laurberg et al. (2008), *Complete factorial sampling* of Donoho & Stodden (2004), and *Separability* of Recht et al. (2012). A necessary and sufficient condition exists as given by the following theorem.

**Theorem 5 (Thomas (1974))** *The decomposition* $\mathbf{X}_0 := \mathbf{W}_0 \mathbf{H}_0$ *is unique up to permutation and positive scaling of columns (resp. rows) of* $\mathbf{W}_0$ *(resp.* $\mathbf{H}_0$*) if and only if the $K$-dimensional positive orthant is the only $K$-simplicial cone verifying* $\text{Cone}(\mathbf{W}_0^\top) \subseteq \mathcal{C} \subseteq \text{Cone}(\mathbf{H}_0)$ *where* $\text{Cone}(\mathbf{A})$ *is the cone generated by the rows of* $\mathbf{A}$.

Our first main assumption is:

• **Assumption (A1)** *In the set given by the union of sets:*

$$\{\mathcal{C} \ : \ \text{Cone}(\mathbf{W}_{0\text{train}}^\top) \subseteq \mathcal{C} \subseteq \text{Cone}(\mathbf{H}_0)\} \bigcup \{\mathcal{C} \ : \ \text{Cone}(\mathbf{W}_0^\top) \subseteq \mathcal{C} \subseteq \text{Cone}(\mathbf{H}_{0T})\} \,,$$

*the nonnegative orthant is the only $K$-simplicial cone.*

It is clear that this property is implied by the following one, namely $(\textbf{A'1}) \Rightarrow (\textbf{A1})$.

• **Assumption (A'1)** *In the set*

$$\{\mathcal{C} \ : \ \text{Cone}(\mathbf{W}_{0\text{train}}^\top) \subseteq \mathcal{C} \subseteq \text{Cone}(\mathbf{H}_{0T})\}$$

*the nonnegative orthant is the only $K$-simplicial cone.*

We consider the following standard definition.

**Definition 6 (Javadi & Montanari (2020a))** *The convex hull* $\text{conv}(\mathbf{X}_0)$ *has an internal radius* $\mu > 0$ *if it contains an $K-1$ dimensional ball of radius $\mu$.*

Our second main assumption is that:

• **Assumption (A2)** *Assume that*

$$\text{conv}(\underbrace{\mathbf{T}_{\text{train}}(\mathbf{X}_0)}_{=\mathbf{W}_{0\text{train}}\mathbf{H}_0}) \text{ has internal radius } \mu > 0 \,. \quad\quad \textbf{(A2)}$$

**Theorem 7** *The Assumption* (**A1**) *implies the Property* (**POU**). *Moreover, if* (**A1**) *and* (**A2**) *holds,* $\mathbf{T}(\mathbf{W}\mathbf{H}) = \mathbf{T}(\mathbf{W}_0 \mathbf{H}_0)$ *and* $\mathbf{W}_0 \mathbf{1} = \mathbf{W}\mathbf{1} = \mathbf{1}$ *then* $(\mathbf{W}, \mathbf{H}) = (\mathbf{W}_0, \mathbf{H}_0)$ *up to permutation of columns (resp. rows) of* $\mathbf{W}$ *(resp.* $\mathbf{H}$*), and there is no scaling.*

**Proof.** Proofs are given in Supplement Material. ∎

**Remark 8** *By Theorem 5, observe that (**A'1**) is a necessary and sufficient condition for the uniqueness of the decomposition* $\mathbf{X}_1 = \mathbf{W}_{0\text{train}} \mathbf{H}_{0T}$. *Then, using* $(\textbf{A'1}) \Rightarrow (\textbf{A1})$, *we understand that if decomposition of* $\mathbf{X}_1 = \mathbf{W}_{0\text{train}} \mathbf{H}_{0T}$ *is unique then* (**POU**) *holds.*

### 1.3.2 Robustness under partial observations

The second issue is *robustness to noise*. To the best of our knowledge, all the results addressing this issue assume that the noise error term is small enough, *e.g.*, Laurberg et al. (2008), Recht et al. (2012), or Javadi & Montanari (2020a). In this paper, we extend these stability result to the nonnegative matrix completion framework (partial observations) and we also assume that noise term $\|\mathbf{F}\|_F$ is small enough.

In the normalized case (*i.e.*, $\mathbf{W1} = \mathbf{1}$), both issues (uniqueness and robustness) can be handle with the notion of $\alpha$-uniqueness, introduced by Javadi & Montanari (2020a). This notion does not handle the matrix completion problem we are addressing. To this end, let us introduce the following notation. Given two matrices $\mathbf{A} \in \mathbb{R}^{n_a \times p}$ and $\mathbf{B} \in \mathbb{R}^{n_b \times p}$ with same row dimension, and $\mathbf{C} \in \mathbb{R}^{n_a \times n_b}$, define the divergence $\mathcal{D}(\mathbf{A}, \mathbf{B})$ as

$$\mathcal{D}(\mathbf{A}, \mathbf{B}) := \min_{\mathcal{C} \geq \mathbf{0}, \ \mathcal{C}\mathbf{1}_{n_b} = \mathbf{1}_{n_a}} \sum_{a=1}^{n_a} \left\| A^{(a)} - \sum_{b=1}^{n_b} C_{ab} B^{(b)} \right\|_F^2,$$
$$= \min_{\mathcal{C} \geq \mathbf{0}, \ \mathcal{C}\mathbf{1}_{n_b} = \mathbf{1}_{n_a}} \|\mathbf{A} - \mathbf{CB}\|_F^2.$$

which is the squared distance between rows of $\mathbf{A}$ and conv($\mathbf{B}$), the convex hull of rows of $\mathbf{B}$. For $\mathbf{B} \in \mathbb{R}^{n \times p}$ define

$$\widetilde{\mathcal{D}}(\mathbf{A}, \mathbf{B}) := \min_{\substack{\mathbf{C} \geq \mathbf{0}, \ \mathbf{C}\mathbf{1}_n = \mathbf{1}_{n_a} \\ \mathbf{T}(\mathbf{N} - \mathbf{B}) = 0}} \|\mathbf{A} - \mathcal{C}\mathbf{N}\|_F^2.$$

**Definition 9 ($\mathbf{T}_\alpha$-unique)** *Given* $\mathbf{X}_0 \in \mathbb{R}^{n \times p}, \mathbf{W}_0 \in \mathbb{R}^{n \times K}, \mathbf{H}_0 \in \mathbb{R}^{K \times p}$, *the factorization* $\mathbf{X}_0 = \mathbf{W}_0\mathbf{H}_0$ *is* $\mathbf{T}_\alpha$-*unique with parameter* $\alpha > 0$ *if for all* $\mathbf{H} \in \mathbb{R}^{K \times p}$ *with* conv($\mathbf{X}_0$) $\subseteq$ conv($\mathbf{H}$):

$$\widetilde{\mathcal{D}}(\mathbf{H}, \mathbf{X}_0)^{1/2} \geq \widetilde{\mathcal{D}}(\mathbf{H}_0, \mathbf{X}_0)^{1/2} + \alpha \left\{ \mathcal{D}(\mathbf{H}, \mathbf{H}_0)^{1/2} + \mathcal{D}(\mathbf{H}_0, \mathbf{H})^{1/2} \right\}.$$

Our third main assumption is given by:

- **Assumption (A3)** *Assume that*

$$\mathbf{X}_0 = \mathbf{W}_0\mathbf{H}_0 \text{ is } \mathbf{T}_\alpha\text{-unique} \tag{A3}$$

**Theorem 10** *If* (A2) *and* (A3) *hold then there exists positive reals* $\Delta$ *and* $\Lambda$ *(depending on* $\mathbf{X}_0$*) such that, for all* $\mathbf{F}$ *such that* $\|\mathbf{F}\|_F \leq \Delta$ *and* $0 \leq \lambda \leq \Lambda$, *any solution* $(\widehat{\mathbf{W}}, \widehat{\mathbf{H}})$ *to* (mAMF) *(if* $\lambda \neq 0$*) or* (mNMF) *(if* $\lambda = 0$*) with observation* (1) *is such that:*

$$\sum_{\ell \leq [K]} \min_{\ell' \leq [K]} \|\mathbf{H}_{0\ell} - \widehat{\mathbf{H}}_{\ell'}\|_2^2 \leq c \|\mathbf{F}\|_F^2,$$

*where* $c$ *is a constant depending only on* $\mathbf{X}_0$.

**Proof.** Proofs are given in Supplement Material. ∎

### 1.4 Outline

The rest of the paper is organized as follows. In Section 2 we discuss the *Sliding Mask Method* (SMM). We present numerical experiments in Section 3, while conclusions are drawn in Section 4. A repository on the numerical experiments can be found at [**link redacted to comply with double blind reviewing**]

## 2 The Sliding Mask Method

### 2.1 Sliding window as forecasting

One is given $N$ time series $\mathbf{M}^{(1)}, \ldots, \mathbf{M}^{(N)} \in \mathbb{R}^T$ over a period of $T$ dates. Recall $\mathbf{M} \in \mathbb{R}^{N \times T}$ is the matrix of observation such that $\mathbf{M}^\top = [(\mathbf{M}^{(1)})^\top \cdots (\mathbf{M}^{(N)})^\top]$ and assumed entry-wise nonnegative. We assume

some *periodicity* in our time series, namely that $\mathbf{M}^\star$ can be split into $B$ matrix blocks of size $N \times P$ where $P = (T + F)/B$, see Figure 1.

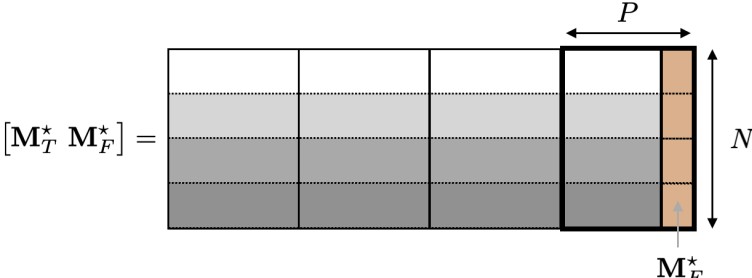

Figure 1: Target matrix $\mathbf{M}^\star$ can be split into $B$ blocks of same time length $P$.

Given $W \geq 1$ and a $T \times N$ matrix $\mathbf{M}$, we define $\mathbf{\Pi}(\mathbf{M})$ the linear operator that piles up $W$ consecutive sub-blocks in a row, as depicted in Figure 2. This process looks at $W$ consecutive blocks in a *sliding* manner. Note that $\mathbf{\Pi}(\mathbf{M})$ is an *incomplete* matrix where the missing values are depicted in orange in Figure 2, they correspond to the time-period to be forecasted. Unless otherwise specified, these unobserved values are set to zero. Remark that $\mathbf{\Pi}(\mathbf{M})$ has $W$ columns blocks, namely $WP$ columns and $(B - W + 1)N$ rows. By an abuse of notation, we also denote

$$\mathbf{\Pi} : \mathbb{R}^{N \times (T+F)} \to \mathbb{R}^{(B-W+1)N \times WP}$$

the same one-to-one linear matrix operation on matrices of size $N \times (T + F)$. In this case, $\mathbf{X}_0 := \mathbf{\Pi}(\mathbf{M}^\star)$ is a *complete* matrix where the orange values have been implemented with the future values of the target $\mathbf{M}_F^\star$.

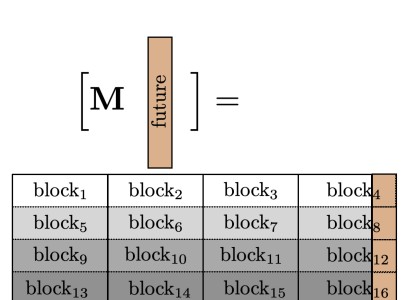

Figure 2: The operator $\mathbf{\Pi}(\mathbf{M})$ outputs an incomplete $(B - W + 1)N \times WP$ matrix given by a mask where the $NF$ orange entries are not observed. These entries corresponds to future times that should be forecasted.

The rationale behind is recasting the forecasting problem as a supervised learning problem where one observes, at each line of $\mathbf{\Pi}(\mathbf{M})$, the $WP - F$ first entries and learn the next $F$ entries. The training set is given by rows $1, 3, 5, 7$ in $\mathbf{\Pi}(\mathbf{M})$ of Figure 2 and the validation set is given by rows $2, 4, 6, 8$ where one aims at predicting the $F$ missing values from the $WP - F$ first values of these rows.

## 2.2 Mask NNMF and Mask AMF

Consider a matrix completion version of NMF with observations

$$\mathbf{X} = \mathbf{\Pi}(\mathbf{M}) = \underbrace{\mathbf{\Pi}(\mathbf{M}_T^\star)}_{\mathbf{X}^\star} + \underbrace{\mathbf{\Pi}(\mathbf{E})}_{\mathbf{F}},$$

and

$$\min_{\mathbf{W}\mathbf{1}=\mathbf{1}, \mathbf{W} \geq 0, \mathbf{H} \geq 0} \|\mathbf{X} - \mathbf{T}(\mathbf{W}\mathbf{H})\|_F^2, \tag{2}$$

where the "mask" operator $\mathbf{T}$ is defined by zeroing the "future" values (in orange in Figure 2). Note that

$$\mathbf{T}(\underbrace{\mathbf{\Pi}(\mathbf{M}^\star)}_{\mathbf{X}_0}) = \mathbf{\Pi}(\mathbf{M}_T^\star) = \mathbf{X}^\star \, .$$

Moreover, note that Problem (2) is equivalent to mask NNMF (mNMF). If we drop the nonnegative constraints on $\mathbf{H}$ and consider the archetypal approach, we obtain mask AMF (mAMF). In particular, Theorem 10 applies proving that (mNMF) and (mAMF) are robust to small noise.

### 2.3 Algorithms

#### 2.3.1 Alternating Least Squares for (mNMF)

The basic algorithmic framework for matrix factorization problems is *Block Coordinate Descent* (BCD) method, which can be straightforwardly adapted to (mNMF) (see Supplement Material). BCD for (mNMF) reduces to *Alternating Least Squares* (ALS) algorithm (see Algorithm 4 in Appendix), when an alternative minimization procedure is performed and matrix $\mathbf{WH}$ is projected onto the linear subspace $\mathbf{T}(\mathbf{N}) = \mathbf{X}$ by means of operator $\mathcal{P}_\mathbf{X}$, as follows:

$$\mathbf{N} := \mathcal{P}_\mathbf{X}(\mathbf{WH}) : \mathbf{T}(\mathbf{N}) = \mathbf{X} \text{ and } \mathbf{T}^\perp \mathbf{N} = \mathbf{WH} \, .$$

*Hierarchical Alternating Least Squares* (HALS) is an ALS-like algorithm obtained by applying an exact coordinate descent method Gillis (2014). Moreover, an accelerated version of HALS is proposed in Gillis & Glineur (2012) (see Supplement Material).

#### 2.3.2 Projected Gradient for (mAMF)

*Proximal Alternative Linear Minimization* (PALM) method, introduced in Bolte et al. (2014) and applied to AMF by Javadi & Montanari (2020a), can be also generalized to (mAMF) (see Algorithm 1).

---

**Algorithm 1** PALM for mAMF

1: **Initialization**: chose $\mathbf{H}^0$, $\mathbf{W}^0 \geq \mathbf{0}$ such that $\mathbf{W}^0 \mathbf{1} = \mathbf{1}$, set $\mathbf{N}^0 := \mathcal{P}_\mathbf{X}(\mathbf{W}^0 \mathbf{H}^0)$ and $i := 0$.
2: **while** stopping criterion is not satisfied **do**
3:     $\widetilde{\mathbf{H}}^i := \mathbf{H}^i - \frac{1}{\gamma_1^i} \mathbf{W}^{i\top} \left( \mathbf{W}^i \mathbf{H}^i - \mathbf{N}^i \right)$              $\triangleright$ Gradient step on $\mathbf{H}$, objective first term
4:     $\mathbf{H}^{i+1} := \widetilde{\mathbf{H}}^i - \frac{\lambda}{\lambda + \gamma_1^i} \left( \widetilde{\mathbf{H}}^i - \mathcal{P}_{\mathrm{conv}(\mathbf{N}^i)}(\widetilde{\mathbf{H}}^i) \right)$    $\triangleright$ Gradient step on $\mathbf{H}$, objective second term
5:     $\mathbf{W}^{i+1} := \mathcal{P}_\Delta \left( \mathbf{W}^i - \frac{1}{\gamma_2^i} \left( \mathbf{W}^i \mathbf{H}^{i+1} - \mathbf{N}^i \right) \mathbf{H}^{i+1\top} \right)$        $\triangleright$ Projected gradient step on $\mathbf{W}$
6:     $\mathbf{N}^{i+1} := \mathcal{P}_\mathbf{X} \left( \mathbf{N}^i + \frac{1}{\gamma_3^i} \left( \mathbf{W}^{i+1} \mathbf{H}^{i+1} - \mathbf{N}^i \right) \right)$          $\triangleright$ Projected gradient step on $\mathbf{N}$
7:     $i := i + 1$
8: **end while**

---

$\mathcal{P}_{\mathrm{conv}(\mathbf{A})}$ is the projection operator onto $\mathrm{conv}(\mathbf{A})$ and $\mathcal{P}_\Delta$ is the projection operator onto the $(N-1)$-dimensional standard simplex $\Delta^N$. The two projections can be efficiently computed by means, *e.g.*, Wolfe algorithm Wolfe (1976) and active set method Condat (2016), respectively.

**Theorem 11** *Let $\varepsilon > 0$. If $\gamma_1^i > \|\mathbf{W}^{i\top} \mathbf{W}^i\|_F$, $\gamma_2^i > \max\left\{ \|\mathbf{H}^{i+1} \mathbf{H}^{i+1\top}\|_F, \varepsilon \right\}$, and $\gamma_3^i > 1$, for each iteration $i$, then the sequence $\left( \mathbf{H}^i, \mathbf{W}^i, \mathbf{N}^i \right)$ generated by Algorithm 1 converges to a stationary point of $\Psi(\mathbf{H}, \mathbf{W}, \mathbf{N}) := f(\mathbf{H}) + g(\mathbf{W}) + p(\mathbf{N}) + h(\mathbf{H}, \mathbf{W}, \mathbf{N})$, where:*

$$f(\mathbf{H}) = \lambda \mathcal{D}(\mathbf{H}, \mathbf{N}) \, , \qquad\qquad g(\mathbf{W}) = \sum_{k=1}^K \mathbb{1}_{\{W_k \in \Delta\}} \, ,$$
$$p(\mathbf{N}) = \mathbb{1}_{\{\mathbf{N} = \mathcal{P}_\mathbf{X}(\mathbf{WH})\}} \, , \qquad\qquad h(\mathbf{H}, \mathbf{W}, \mathbf{N}) = \|\mathbf{N} - \mathbf{WH}\|_F^2 \, .$$

**Proof.** Proof is given in Supplement Material. ■

---

**Algorithm 2** iPALM for mAMF

---

1: **Initialization**: $\mathbf{H}^0$, $\mathbf{W}^0 \geq 0$ such that $\mathbf{W}^0 \mathbf{1} = \mathbf{1}$, set $\mathbf{N}^0 := \mathcal{P}_{\mathbf{X}}(\mathbf{W}^0 \mathbf{H}^0)$, $\mathbf{H}^{-1} := \mathbf{H}^0$, $\mathbf{W}^{-1} := \mathbf{W}^0$,
$\mathbf{N}^{-1} := \mathbf{N}^0$, and $i := 0$.
2: **while** stopping criterion is not satisfied **do**
3:      $\mathbf{H}_1^i := \mathbf{H}^i + \alpha_1^i \left( \mathbf{H}^i - \mathbf{H}^{i-1} \right)$, $\mathbf{H}_2^i := \mathbf{H}^i + \beta_1^i \left( \mathbf{H}^i - \mathbf{H}^{i-1} \right)$          ▷ Inertial $\mathbf{H}$
4:      $\widetilde{\mathbf{H}}^i := \mathbf{H}_1^i - \frac{1}{\gamma_1^i} \mathbf{W}^{i^\top} \left( \mathbf{H}_2^i \mathbf{W}^i - \mathbf{N}^i \right)$          ▷ Gradient step on $\mathbf{H}$, objective first term
5:      $\mathbf{H}^{i+1} := \widetilde{\mathbf{H}}^i - \frac{\lambda}{\lambda + \gamma_1^i} \left( \widetilde{\mathbf{H}}^i - \mathcal{P}_{\mathrm{conv}(\mathbf{N}^i)}(\widetilde{\mathbf{H}}^i) \right)$          ▷ Gradient step on $\mathbf{H}$, objective second term
6:      $\mathbf{W}_1^i := \mathbf{W}^i + \alpha_2^i \left( \mathbf{W}^i - \mathbf{W}^{i-1} \right)$, $\mathbf{W}_2^i := \mathbf{W}_1^i + \beta_2^i \left( \mathbf{W}^i - \mathbf{W}^{i-1} \right)$          ▷ Inertial $\mathbf{W}$
7:      $\mathbf{W}^{i+1} := \mathcal{P}_\Delta \left( \mathbf{W}_1^i - \frac{1}{\gamma_2^i} \left( \mathbf{W}_2^i \mathbf{H}^{i+1} - N^i \right) \mathbf{H}^{i+1^\top} \right)$          ▷ Projected gradient step on $\mathbf{W}$
8:      $\mathbf{N}_1^i := \mathbf{N}_1^i + \alpha_3^i \left( \mathbf{N}^i - \mathbf{N}^{i-1} \right)$, $\mathbf{N}_2^i := \mathbf{N}_1^i + \beta_3^i \left( \mathbf{N}^i - \mathbf{N}^{i-1} \right)$          ▷ Inertial $\mathbf{N}$
9:      $\mathbf{N}^{i+1} := \mathcal{P}_{\mathbf{X}} \left( \mathbf{N}_1^i + \frac{1}{\gamma_3^i} \left( \mathbf{W}^{i+1} \mathbf{H}^{i+1} - \mathbf{N}_2^i \right) \right)$          ▷ Projected gradient step on $\mathbf{N}$
10:      $i := i + 1$
11: **end while**

---

Finally, the inertial PALM (iPALM) method, introduced for NMF in Pock & Sabach (2016), is generalized to (mAMF) in Algorithm 2.

**Remark 12** *If, for all iterations $i$, $\alpha_1^i = \alpha_2^i = 0$ and $\beta_1^i = \beta_2^i = 0$, iPALM reduces to PALM.*

### 2.3.3 Stopping criterion for normalized NMF

For NNMF, KKT conditions regarding matrix $\mathbf{W}$ are the following (see Supplement Material):

$$\mathbf{W} \circ \left( (\mathbf{W}\mathbf{H} - \mathbf{N})\mathbf{H}^\top + t\,\mathbf{1}_K^\top \right) = 0\,.$$

By complementary condition, it follows that, $\forall j$, $t_i = ((\mathbf{W}\mathbf{H} - \mathbf{N})\mathbf{H}^\top)_{i,j}$. Hence, we compute $t_i$ by selecting, for each row $W^{(i)}$, any positive entry $W_{i,j} > 0$.

**Remark 13** *Numerically to obtain a robust estimation of $t_i$, we can average the corresponding values calculated per entry $W_{i,j}$.*

Let $\varepsilon_{\mathbf{W}}$, $\varepsilon_{\mathbf{H}}$, and $\varepsilon_{\mathbf{R}}$ be three positive thresholds. The stopping criterion for the previous algorithms consists in a combination of:

1. the maximum number of iterations;

2. the Frobenius norm of the difference of $\mathbf{W}$ and $\mathbf{H}$ at two consecutive iterations, *i.e.*, the algorithm stops if $\|\mathbf{W}^{i+1} - \mathbf{W}^i\|_F \leq \varepsilon_{\mathbf{W}}\ \wedge\ \|\mathbf{H}^{i+1} - \mathbf{H}^i\|_F \leq \varepsilon_{\mathbf{H}}$;

3. a novel criterion based on KKT condition, *i.e.*, the algorithm stops if it holds that $\|\mathbf{R}(\mathbf{W}^{i+1})\|_F + \|\mathbf{R}(\mathbf{H}^{i+1})\|_F \leq \varepsilon_{\mathbf{R}}$ where matrices $\mathbf{R}(\mathbf{W})$ and $\mathbf{R}(\mathbf{H})$ are defined as $\mathbf{R}(\mathbf{W})_{i,j} := |(\mathbf{W}\mathbf{H} - \mathbf{N})\mathbf{H}^\top)_{i,j} + t_i|\mathbb{1}_{\{W_{i,j} \neq 0\}}$ and $\mathbf{R}(\mathbf{H})_{i,j} := |\mathbf{W}^\top(\mathbf{W}\mathbf{H} - \mathbf{N}))_{i,j}|\mathbb{1}_{\{H_{i,j} \neq 0\}}$, respectively.

### 2.4 Large-scale data-set

Assume the observed matrix $\mathbf{X} = \mathbf{\Pi}(\mathbf{M})$ is large scaled, namely one has to forecast a large number $N$ of times series (*e.g.* more than $100,000$) and possibly a large number of time stamps $T$. The strategy, described in Section 1.3.1 in Cichocki et al. (2009) for NMF, is to learn the $\mathbf{H} \in \mathbb{R}^{K \times T}$ matrix from a sub-matrix $\mathbf{N}_r \in \mathbb{R}^{r \times T}$ of $K \leq r \ll N$ rows of $\mathbf{N} \in \mathbb{R}^{n \times T}$, and learn the $\mathbf{W} \in \mathbb{R}^{N \times K}$ matrix from a sub-matrix $\mathbf{N}_c \in \mathbb{R}^{N \times c}$ of $K \leq c \ll T$ columns of $\mathbf{N} \in \mathbb{R}^{N \times T}$. We denote by $\mathbf{H}_c$ the sub-matrix of $\mathbf{H}$ given by the columns appearing in $\mathbf{N}_c$ and $\mathbf{W}_r$ the sub-matrix of $\mathbf{H}$ given by the columns appearing in $\mathbf{N}_c$.

This strategy can be generalized to (mNMF) and (mAMF). For (mNMF) this generalization is straightforward, and for (mAMF) one need to change Steps 3-5 in Algorithm 1 as follows:

$$\widetilde{\mathbf{H}}^i := \mathbf{H}^i - \frac{1}{\gamma_1^i}(\mathbf{W}_r^i)^\top \left(\mathbf{W}_r^i \mathbf{H}^i - \mathbf{N}_r^i\right)$$

$$\mathbf{H}^{i+1} := \widetilde{\mathbf{H}}^i - \frac{\lambda}{\lambda + \gamma_1^i}\left(\widetilde{\mathbf{H}}^i - \mathcal{P}_{\mathrm{conv}(\mathbf{N}^i)}(\tilde{\mathbf{H}}^i)\right)$$

$$\mathbf{W}^{i+1} := \mathcal{P}_\Delta \left(\mathbf{W}^i - \frac{1}{\gamma_2^i}\left(\mathbf{W}^i \mathbf{H}_c^{i+1} - \mathbf{N}_c^i\right)(\mathbf{H}_c^{i+1})^\top\right).$$

Same approach is exploited for Algorithm 2.

## 3 Numerical Experiments

We tested SMM with random initialization of matrices $\mathbf{H}^0, \mathbf{W}^0$. Each entry in $\mathbf{H}^0$ is randomly selected in $[0, h]$ where $h > 0$ is chosen by practitioner. Each row of matrix $\mathbf{W}^0$ is randomly generated in the corresponding standard simplex.

For SMM we implemented both HALS for (mNMF) and PALM for (mAMF). Moreover, we consider two different strategies to define matrix $\mathbf{\Pi}(\mathbf{M})$: with *non-overlapping* (mAMF and mNMF) and *overlapping* sliding intervals (mAMFo and mNMFo). In the overlapping strategy, we replicate the half of each sub-block in which the matrix $\mathbf{M}$ is sub-divided.

Moreover, we have compared our method with other classically-designed mainstream time series forecasting methods such as *Random Forest Regression* (RFR) and *EXponential Smoothing* (EXS), *Long Short-Term Memory* (LSTM) and *Gated Recurrent Units* (GRU) deep neural networks with preliminary data standardization Shewalkar et al. (2019), and *Seasonal Auto-Regressive Integrated Moving Average with eXogenous factors* (SARIMAX) models Douc et al. (2014). In our computational experiments, we consider hundreds of time series, so we do not benchmark against time series transformer model, which are suitably designed instead for huge-scale time series forecasting problems.

The quality of the forecasted matrix $\mathbf{M}_F$ is measured by the relative root-mean-squared error (RRMSE) and the relative mean-percentage error (RMPE):

$$\mathrm{RRMSE} = \frac{\|\mathbf{M}_F - \mathbf{M}_F^\star\|_F}{\|\mathbf{M}_F^\star\|_F}, \ \mathrm{RMPE} = \frac{\|\mathbf{M}_F - \mathbf{M}_F^\star\|_1}{\|\mathbf{M}_F^\star\|_1}.$$

The interested reader may find a github repository on numerical experiments at [**link redacted to comply with double blind reviewing**]

We run all the numerical tests on a MacBook Pro mounting macOS Ventura 13.6.1 with Apple M2 chip and 8 GB LPDDR5 memory RAM.

### 3.1 Real-world data-sets

The numerical experiments refer to the following real-world data-sets:

- weekly and daily electricity consumption data-sets of 370 Portuguese customers during the period 2011-2014, Trindade (2015)

- twin gas measurements data-set of five replicates of an 8-MOX gas sensor, Fonollosa (2016)

- Istanbul Stock Exchange returns with seven other international index for the period 2009-2011, Akbilgic (2013)

- demand forecasting orders in a Brazilian logistics company collected for 60 days, Ferreira et al. (2017)

- daily electricity transformer temperature (ETT) measurements, Zhou et al. (2020)

Tables 1-2 (see Appendix) report the cross-validated RRMSE and RMPE on observed values obtained during the computational tests for each method. Table 5 reports the CPU time for the forecasting phase. We highlight best results in **bold**, and second best results are underlined.

Our method is always the best or the second best one among all the approaches for all the data-set we tested in terms of RRMSE and RMPE indices, and there is no other method performing better. Concerning the CPU times, our novel methods are competitive against EXP, being the fastest or the second fastest ones.

mAMFo seems to be the most promising algorithm in terms of efficacy for the first five data-sets, while mAMF and mNMF are the best methods for the last four ETT data-sets. The ETT data-sets are characterized by a pronounced periodicity: in this case the plain methods performs quite well; while the first five data-sets are less markedly periodic and in this case the overlap versions aims to introduce a sort of artificial periodicity in the dataset by replicating portions of the sub-block in which each row of **M** is split.

### 3.2 Synthetic data-sets

Further computational experiments have been realized by considering additional synthetic data-sets. In particular, we generated three data sets by replicating $1,000$ short time series (with 10 time periods) 10 times and adding white noise multiplied by a constant factor $\sigma$ to each time series entry separately. We choose $\sigma \in \{0.005, 0.1, 1\}$. We refer to the these data-sets as "low noise", "medium noise", and "high noise", respectively.

An additional synthetic data-set has been generated by considering few probability vectors, and computing the entire matrix **W** by randomly choosing a probability vector and adding white noise. A completely randomly generated matrix **H** is multiplied to **W** to obtain the whole matrix $\mathbf{M}^* := \mathbf{WH}$. We refer to this dataset as "few distribution".

Finally, the last synthetic data-set is obtained by generating matrix **H** by replicating a small time series (with 50 time periods) 100 times and adding white noise multiplied by a constant factor $\sigma = 1$ and matrix **W** of suitable dimensions, whose rows are uniformly distributed over the corresponding dimensional simplex. Then, we set matrix $\mathbf{M}^* := \mathbf{WH}$. We refer to this last dataset as "periodic archetypes".

Tables 3-4 and 6 (see Appendix) report the cross-validated RRMSE and RMPE indices, and the CPU time for the forecasting phase, respectively, referring to synthetic generated datasets. The more pronounced the periodicity of the time series or of the archetypes, the better the performances of our proposed NMF-like methods: in this case, the more realistic the hypothesis that the whole data-set can be expressed as convex combinations of few archetypes, having a low-rank representation. We are able to replicate SARIMAX performances in a much shorter time; while, for the last two synthetic data-sets, mAMF and mNMF outperform the benchmarks.

## 4 Conclusions and Perspectives

In this paper, we have introduced and described a novel approach for the time series forecasting problem relying on nonnegative matrix factorization. We apply this algorithm to realistic data-sets and synthetics data-sets, showing the forecasting capabilities of the proposed methodology.

Moreover, we have shown several uniqueness and robustness theoretical results for the solution of the matrix factorization problems faced by the proposed algorithm, namely the *Sliding Mask Method*.

The strength of the proposed methodology consists in its relatively loose assumptions, mainly by supposing that time series matrix can be efficiently described by a low rank nonnegative decomposition, and that the time series are periodic for the *Sliding Mask Method*. Moreover, the *Sliding Mask Method* can be applied in presence of missing entries in the dataset: in this latter case, in the mask operation one should consider only the known past values of each time series.

Future works consists in embedding side information in the forecasting procedure by extending algorithms in Mei et al. (2019) to the *Sliding Mask Method.*

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

| Algorithms | RFR | | LSTM | | GRU | | EXP | | SARIMAX | |
|---|---|---|---|---|---|---|---|---|---|---|
| Metrics | RRMSE | RMPE | RRMSE | RMPE | RRMSE | RMPE | RRMSE | RMPE | RRMSE | RMPE |
| weekly electricity | **7.25%** | 8.61% | 27.85% | 15.64% | 26.04% | 15.92% | 10.07% | 7.98% | 9.05% | **7.42%** |
| daily electricity | 12.16% | 47.78% | 12.42% | 46.49% | 12.03% | 45.90% | 11.25% | 43.83% | **9.85%** | 43.16% |
| gas | 66.80% | 71.61% | 62.97% | 68.38% | 62.87% | 67.90% | 63.35% | 68.16% | 45.58% | 52.83% |
| Istanbul | 15.37% | 18.32% | 16.22% | 20.96% | 20.01% | 26.87% | 15.46% | 18.64% | 14.75% | 17.01% |
| demand orders | **16.64%** | **25.12%** | 18.47% | 29.45% | 18.89% | 30.60% | 18.34% | 25.94% | 22.99% | 31.42% |
| ETTh1 | 12.96% | 17.98% | 14.86% | 18.78% | 14.71% | 18.85% | 12.37% | **13.65%** | 13.36% | 15.94% |
| ETTh2 | 6.47% | 7.60% | 14.17% | 13.75% | 14.44% | 14.36% | 14.06% | 13.67% | 12.76% | 13.03% |
| ETTm1 | 12.81% | 17.42% | 13.39% | 17.96% | 14.13% | 18.63% | 11.45% | 14.20% | 12.29% | 16.45% |
| ETTm2 | 5.81% | **7.16%** | 14.29% | 13.89% | 14.46% | 14.03% | 13.18% | 12.88% | 13.16% | 12.95% |

Table 1: Real-world data-sets: RRMSE and RPME indices (benchmarks)

| Algorithms | mAMF | | mNMF | | mAMFo | | mNMFo | |
|---|---|---|---|---|---|---|---|---|
| Metrics | RRMSE | RMPE | RRMSE | RMPE | RRMSE | RMPE | RRMSE | RMPE |
| weekly electricity | 12.74% | 23.45% | 9.34% | 16.76% | 8.13% | 12.16% | 12.56% | 13.93% |
| daily electricity | 11.68% | 37.20% | 14.44% | 40.51% | 12.08% | 35.92% | 18.36% | **32.34%** |
| gas | 50.93% | 46.41% | 65.41% | 65.40% | **48.33%** | 49.86% | 56.54% | 58.62% |
| Istanbul | 15.90% | 18.49% | 17.98% | 17.74% | **14.74%** | **15.46%** | 18.02% | 18.83% |
| demand orders | 23.02% | 34.72% | 24.83% | 34.95% | 28.25% | 39.88% | 17.68% | 31.85% |
| ETTh1 | **11.72%** | 14.71% | 11.93% | 15.22% | 14.81% | 24.07% | 23.86% | 25.57% |
| ETTh2 | 5.90% | 7.39% | **4.39%** | **6.68%** | 21.86% | 23.03% | 18.29% | 21.29% |
| ETTm1 | **10.63%** | 14.82% | 11.26% | **13.49%** | 15.27% | 26.23% | 20.02% | 26.30% |
| ETTm2 | 5.83% | 7.28% | **5.04%** | 7.53% | 20.82% | 22.35% | 16.90% | 21.41% |

Table 2: Real-world data-sets: RRMSE and RPME indices (novel algorithms)

| Algorithms | RFR | | LSTM | | GRU | | EXP | | SARIMAX | |
|---|---|---|---|---|---|---|---|---|---|---|
| Metrics | RRMSE | RMPE | RRMSE | RMPE | RRMSE | RMPE | RRMSE | RMPE | RRMSE | RMPE |
| low noise | 8.65% | 22.76% | 16.52% | 46.03% | 16.76% | 46.46% | 16.97% | 48.02% | **0.19%** | **0.33%** |
| medium noise | 8.42% | 21.95% | 15.66% | 42.68% | 15.67% | 42.98% | 15.94% | 44.25% | **1.97%** | **4.94%** |
| high noise | 11.73% | 30.26% | 13.03% | 33.67% | 12.97% | 33.47% | 13.02% | 33.27% | 15.24% | **27.37%** |
| few distributions | 5.79% | 9.44% | 6.76% | 11.87% | 6.77% | 11.89% | 5.73% | 9.32% | 5.76% | 8.81% |
| periodic arch. | 17.12% | 20.24% | 21.55% | 29.04% | 21.48% | 28.94% | 21.09% | 25.87% | 28.41% | 35.53% |

Table 3: Synthetic data-sets: RRMSE and RMPE indices (benchmarks)

| Algorithms | mAMF | | mNMF | | mAMFo | | mNMFo | |
|---|---|---|---|---|---|---|---|---|
| Metrics | RRMSE | RMPE | RRMSE | RMPE | RRMSE | RMPE | RRMSE | RMPE |
| low noise | 0.61% | 1.02% | 1.34% | 1.18% | 0.60% | 1.03% | 1.50% | 1.61% |
| medium noise | 2.36% | 5.35% | 2.72% | 5.84% | 2.42% | 5.81% | 2.45% | 5.67% |
| high noise | 11.90% | 28.40% | 13.03% | 30.83% | 11.57% | 28.50% | **11.18%** | 27.53% |
| few distributions | **4.72%** | **6.69%** | 8.67% | 15.00% | 5.58% | 7.61% | 6.29% | 9.09% |
| periodic arch. | 16.42% | 18.74% | **3.32%** | **3.82%** | 25.91% | 32.80% | 24.97% | 31.01% |

Table 4: Synthetic data-sets: RRMSE and RMPE indices (novel algorithms)

| Algorithms | mAMF | mNMF | mAMFo | mNMFo | RFR | LSTM | GRU | EXP | SARIMAX |
|---|---|---|---|---|---|---|---|---|---|
| weekly electricity | 1.48 | 1.23 | 1.86 | **0.18** | 16.63 | 286.45 | 340.53 | 0.71 | 98.22 |
| daily electricity | 3.67 | **0.64** | 5.32 | 1.05 | 125.70 | 739.06 | 691.44 | 1.07 | 840.13 |
| gas | 0.32 | 0.18 | 3.77 | **0.13** | 18.05 | 330.49 | 466.98 | 0.89 | 90.18 |
| Istanbul | 0.03 | <0.01 | 0.15 | <0.01 | 0.39 | 6.62 | 6.35 | 0.01 | 4.25 |
| demand orders | 0.02 | 0.06 | 0.02 | 0.07 | 0.48 | 8.09 | 7.96 | **0.01** | 1.02 |
| ETTh1 | 0.06 | 0.02 | 0.06 | **0.01** | 0.02 | 6.02 | 6.31 | **0.01** | 6.81 |
| ETTh2 | 0.04 | 0.06 | 0.08 | 0.06 | 0.53 | 6.30 | 6.79 | **0.01** | 9.51 |
| ETTm1 | 0.08 | 0.03 | 0.06 | 0.11 | 0.03 | 6.02 | 6.04 | **0.01** | 6.51 |
| ETTm2 | 0.04 | 0.05 | 0.07 | 0.06 | 0.05 | 6.21 | 6.65 | **0.01** | 9.63 |

Table 5: Real-world data-sets: CPU times (seconds)

| Algorithms | mAMF | mNMF | mAMFo | mNMFo | RFR | LSTM | GRU | EXP | SARIMAX |
|---|---|---|---|---|---|---|---|---|---|
| low noise | 2.16 | **1.15** | 2.30 | 1.20 | 45.53 | 1013.18 | 2739.64 | 5.01 | 213.23 |
| medium noise | 1.91 | 0.96 | 1.93 | **0.95** | 45.11 | 1031.13 | 2752.62 | 5.11 | 317.79 |
| high noise | 1.69 | **0.44** | 1.89 | 0.53 | 45.66 | 1026.00 | 2763.16 | 5.09 | 265.02 |
| periodic arch. | 1.56 | **0.45** | 4.52 | 1.49 | 59.07 | 1056.76 | 2610.67 | 6.02 | 1013.43 |
| few distributions | **0.56** | 0.60 | 6.67 | 1.63 | 60.89 | 1097.85 | 2674.51 | 6.02 | 1026.83 |

Table 6: Synthetic data-sets: CPU times (seconds)

# A Variants of Nonnegative Matrix Factorization problems

| Acronym | Name | Objective | Constraints: $\mathbf{W} \geq \mathbf{0}$ + |
|---------|------|-----------|------------------------------------------------|
| NMF | Nonnegative Matrix Factorization Cichocki & Zdunek (2006) | $\mathbf{F}_1$ | $\mathbf{H} \geq \mathbf{0}$ |
| SNMF | Semi NMF Gillis & Kumarg (2015) | $\mathbf{F}_1$ | |
| NNMF | Normalized NMF | $\mathbf{F}_1$ | $\mathbf{H} \geq \mathbf{0}, \mathbf{W1} = \mathbf{1}$ |
| SNNMF | Semi Normalized NMF | $\mathbf{F}_1$ | $\mathbf{W1} = \mathbf{1}$ |
| AMF | Archetypal Matrix Factorization Javadi & Montanari (2020a) | $\mathbf{F}_2$ | $\mathbf{W1} = \mathbf{1}, \mathbf{V} \geq \mathbf{0}, \mathbf{V1} = \mathbf{1}$ |
| ANMF | Archetypal NMF | $\mathbf{F}_2$ | $\mathbf{H} \geq \mathbf{0}, \mathbf{V} \geq \mathbf{0}, \mathbf{V1} = \mathbf{1}$ |
| ANNMF | Archetypal Normalized NMF | $\mathbf{F}_2$ | $\mathbf{W1} = \mathbf{1}, \mathbf{H} \geq \mathbf{0}, \mathbf{V} \geq \mathbf{0}, \mathbf{V1} = \mathbf{1}$ |
| mNMF | Mask NNMF | $\mathbf{F}_3$ | $\mathbf{T}(\mathbf{N}) = \mathbf{X}, \mathbf{W1} = \mathbf{1}, \mathbf{H} \geq \mathbf{0}$ |
| mAMF | Mask AMF | $\mathbf{F}_4$ | $\mathbf{T}(\mathbf{N}) = \mathbf{X}, \mathbf{W1} = \mathbf{1}, \mathbf{V} \geq \mathbf{0}, \mathbf{V1} = \mathbf{1}$ |

Table 7: The seven block convex programs achieving matrix factorization of nonnegative matrices. The objectives are $\mathbf{F}_1 := \|\mathbf{M} - \mathbf{WH}\|_F^2$ and $\mathbf{F}_2 := \|\mathbf{M} - \mathbf{WH}\|_F^2 + \lambda\|\mathbf{H} - \mathbf{VM}\|_F^2$. The two last lines are SMM procedures with sliding operator $\mathbf{\Pi}$ and objectives $\mathbf{F}_3 := \|\mathbf{N} - \mathbf{WH}\|_F^2$ and $\mathbf{F}_4 := \|\mathbf{N} - \mathbf{WH}\|_F^2 + \lambda\|\mathbf{H} - \mathbf{VN}\|_F^2$.

# B Proofs

## B.1 Proof of Theorem 7

• We start by proving that *Condition* (**A1**) *is sufficient for* (**POU**).

Let $\mathbf{H}_0 := [\mathbf{H}_{0T} \ \mathbf{H}_{0F}]$, $\mathbf{W}_0^\top := [\mathbf{W}_{0\text{train}}^\top \ \mathbf{W}_{0\text{test}}^\top]$, $\mathbf{H} := [\mathbf{H}_T \ \mathbf{H}_F]$, and $\mathbf{W}^\top := [\mathbf{W}_{\text{train}}^\top \ \mathbf{W}_{\text{test}}^\top]$. Assumption (**A1**) implies that decomposition $\mathbf{W}_{0\text{train}}\mathbf{H}_0$ and $\mathbf{W}_0\mathbf{H}_{0T}$ are unique. By Theorem 1, it holds

$$\mathbf{W}_{0\text{train}}\mathbf{H}_0 = \mathbf{W}_{\text{train}}\mathbf{H} \Longrightarrow (\mathbf{W}_{0\text{train}}, \mathbf{H}_0) \equiv (\mathbf{W}_{\text{train}}, \mathbf{H})$$
$$\mathbf{W}_0\mathbf{H}_{0T} = \mathbf{W}\mathbf{H}_T \Longrightarrow (\mathbf{W}_0, \mathbf{H}_{0T}) \equiv (\mathbf{W}, \mathbf{H}_T),$$

where $\equiv$ stands for equality up to permutation and positive scaling of columns (resp. rows) of $\mathbf{W}_0$ (resp. $\mathbf{H}_0$). Hence, if (**A1**) holds, then

$$(\mathbf{W}_{0\text{train}}\mathbf{H}_0 = \mathbf{W}_{\text{train}}\mathbf{H}) \wedge (\mathbf{W}_0\mathbf{H}_{0T} = \mathbf{W}\mathbf{H}_T) \Longrightarrow (\mathbf{W}_0, \mathbf{H}_0) \equiv (\mathbf{W}, \mathbf{H}). \quad (3)$$

Moreover, note $\mathbf{T}(\mathbf{W}_{0\text{train}}\mathbf{H}_0) = \mathbf{T}_{\text{train}}(\mathbf{X}_0) = \mathbf{W}_{0\text{train}}\mathbf{H}_0$ and $\mathbf{T}(\mathbf{W}_0\mathbf{H}_{0T}) = \mathbf{T}_T(\mathbf{X}_0) = \mathbf{W}_0\mathbf{H}_{0T}$ (same equations holds for $(\mathbf{W}, \mathbf{H})$). We deduce that $\mathbf{T}(\mathbf{W}_0\mathbf{H}_0) = \mathbf{T}(\mathbf{W}\mathbf{H})$ implies $(\mathbf{W}_{0\text{train}}\mathbf{H}_0 = \mathbf{W}_{\text{train}}\mathbf{H}) \wedge (\mathbf{W}_0\mathbf{H}_{0T} = \mathbf{W}\mathbf{H}_T)$. We deduce the result by (3).

• We prove that *If* (**A1**) *and* (**A2**) *holds,* $\mathbf{T}(\mathbf{W}\mathbf{H}) = \mathbf{T}(\mathbf{W}_0\mathbf{H}_0)$ *and* $\mathbf{W}_0\mathbf{1} = \mathbf{W1} = \mathbf{1}$ *then* $(\mathbf{W}, \mathbf{H}) = (\mathbf{W}_0, \mathbf{H}_0)$ *up to permutation of columns (resp. rows) of* $\mathbf{W}$ *(resp.* $\mathbf{H}$*), and there is no scaling.*

By the previous point, we now that (**A1**) implies $(\mathbf{W}_0, \mathbf{H}_0) \equiv (\mathbf{W}, \mathbf{H})$. So that there exist $\lambda_1, \ldots, \lambda_K$ positive and a permutation $\sigma(1), \ldots, \sigma(K)$ such that

$$\forall i \in [n - N], \forall k \in [K], \quad (\mathbf{W})_k^{(i)} = \lambda_{\sigma(k)}(\mathbf{W}_0)_{\sigma(k)}^{(i)}.$$

Recall that $\mathbf{W1} = \mathbf{1}$ (resp. $\mathbf{W}_0\mathbf{1} = \mathbf{1}$) so that the rows of $\mathbf{W}$ (resp. $\mathbf{W}_0$) belongs to the affine space

$$\mathcal{A}_\mathbf{1} := \left\{ w \in \mathbb{R}^K \ : \ \langle w, \mathbf{1} \rangle = 1 \right\}.$$

Namely, for a given row $i \in [n - N]$, we have

$$(\mathbf{W}_0)^{(i)} \mathbf{1} = \mathbf{1} \Rightarrow \sum_{k=1}^{K} (\mathbf{W}_0)_k^{(i)} = 1$$

$$\mathbf{W}^{(i)} \mathbf{1} = \mathbf{1} \Rightarrow \sum_{k=1}^{K} \lambda_{\sigma(k)} (\mathbf{W}_0)_{\sigma(k)}^{(i)} = 1$$

Which proves that $(\mathbf{W}_0)^{(i)} \in \mathcal{A}_{\mathbf{1}} \cap \mathcal{A}_{\lambda_{\sigma^{-1}}}$, for all $i \in [n - N]$, where

$$\mathcal{A}_{\lambda_{\sigma^{-1}}} := \left\{ w \in \mathbb{R}^K \ : \ \sum_{k=1}^{K} \lambda_{\sigma^{-1}(k)} w_k = 1 \right\},$$

is the affine space orthogonal to $\mathbf{d} := (\lambda_{\sigma^{-1}(1)}, \ldots, \lambda_{\sigma^{-1}(K)})$. We deduce that the rows $(\mathbf{W}_0)^{(i)}$ belong to the affine space

$$\mathcal{A} := \left\{ w \in \mathbb{R}^K \ : \ \langle w, \mathbf{1} \rangle = 1 \text{ and } \langle w, \mathbf{d} \rangle = 1 \right\}$$

which is of:
• co-dimension 2 if $\mathbf{d}$ is not proportional to $\mathbf{1}$;
• co-dimension 1 if there exists $\lambda > 0$ such that $\mathbf{d} = \lambda \mathbf{1}$. In this latter case, $\lambda = 1$ and for all $k \in [K]$, $\lambda_k = 1$, namely there is no scaling of the columns.

If $\mathcal{A}$ is of co-dimension 2 then $\mathcal{A}$ is of dimension $K - 2$ and $\mathrm{Conv}(\mathbf{W}_{0,\mathrm{train}}) \subseteq \mathcal{A}$ cannot contain a ball of dimension $K - 1$, which implies that $\mathrm{Conv}(\mathbf{T}_{\mathrm{train}}(\mathbf{X}_0)) \subseteq \mathcal{A} \times H$ is of dimension at most $K - 2$ and it cannot contain a ball of dimension $K - 1$ (i.e., co-dimension 1), where $\mathcal{A} \times H = \{x \ : \ \exists a \in \mathcal{A} \text{ s.t. } x = a^\top H\}$. This latter is a contradiction under (**A2**). We deduce that $\mathcal{A}$ is of co-dimension 2, and so there is no scaling.

## B.2 Proof of Theorem 10

This proof follows the pioneering work Javadi & Montanari (2020a). In this latter paper, the authors consider neither masks $\mathbf{T}$ nor nonnegative constraints on $\mathbf{H}$ as in (mNMF). Nevertheless,
1/ considering the hard constrained programs (4) and (6) below;
2/ remarking that it holds $\widetilde{\mathcal{D}}(\mathbf{H}, \mathbf{X}) \leq \mathcal{D}(\mathbf{H}, \mathbf{X})$ and $\overline{\mathcal{D}}(\mathbf{X}, \mathbf{H}) \leq \mathcal{D}(\mathbf{X}, \mathbf{H})$;
then a careful reader can note that their proof extends to masks $\mathbf{T}$ and nonnegative constraints on $\mathbf{H}$. For sake of completeness we reproduce here the steps that need to be changed in their proof. A reading guide of the 60 pages proof of Javadi & Montanari (2020b) is given in Section C.

**Step 1: reduction to hard constrained Programs (4) and (6)**

Consider the constrained problem:

$$\widehat{\mathbf{H}} \in \arg\min_{\mathbf{H}} \ \widetilde{\mathcal{D}}(\mathbf{H}, \mathbf{X})$$
$$\text{s.t. } \overline{\mathcal{D}}(\mathbf{X}, \mathbf{H}) \leq \Delta_1^2. \tag{4}$$

where

$$\overline{\mathcal{D}}(\mathbf{X}, \mathbf{H}) := \min_{\mathbf{W} \geq \mathbf{0}, \ \mathbf{W}\mathbf{1}=\mathbf{1}} \| \mathbf{T}(\mathbf{X} - \mathbf{W}\mathbf{H}) \|_F^2$$

Then (mAMF) can be seen as Lagrangian formulation of this problem setting $\Delta_1^2 = \overline{\mathcal{D}}(\mathbf{X}, \widehat{\mathbf{H}}_{(\mathrm{mAMF})})$, where $\widehat{\mathbf{H}}_{(\mathrm{mAMF})}$ is a solution to (mAMF). We choose $\Delta_1$ so as to bound the noise level $\|\mathbf{F}\|_F$

$$\Delta_1^2 \geq \|\mathbf{F}\|_F^2. \tag{5}$$

Consider the constrained problem:

$$\widehat{\mathbf{H}} \in \arg\min_{\mathbf{H} \geq \mathbf{0}} \ \widetilde{\mathcal{D}}(\mathbf{H}, \mathbf{X})$$
$$\text{s.t. } \overline{\mathcal{D}}(\mathbf{X}, \mathbf{H}) \leq \Delta_2^2. \tag{6}$$

Then (mNMF) can be seen as Lagrangian formulation of this problem setting $\Delta_2^2 = \overline{\mathcal{D}}(\mathbf{X}, \widehat{\mathbf{H}}_{\text{(mNMF)}})$, where $\widehat{\mathbf{H}}_{\text{(mNMF)}}$ is a solution to (mNMF). We choose $\Delta_1$ so as to bound the noise level $\|\mathbf{F}\|_F$

$$\Delta_2^2 \geq \|\mathbf{F}\|_F^2. \tag{7}$$

**Step 2: First bound on the loss**

Denote $\mathcal{D} := \left\{\mathcal{D}(\mathbf{H}, \mathbf{H}_0)^{1/2} + \mathcal{D}(\mathbf{H}_0, \mathbf{H})^{1/2}\right\}$. By Assumption (**A2**) we have

$$\boldsymbol{z}_0 + \boldsymbol{U}B_{K-1}(\mu) \subseteq \text{conv}(\boldsymbol{X}_0) \subseteq \text{conv}(\boldsymbol{H}_0),$$

where $\boldsymbol{z}_0 + \boldsymbol{U}B_{K-1}(\mu)$ is a parametrization of the ball of center $\boldsymbol{z}_0$ and radius $\mu$ described in Assumption (**A2**) with $\boldsymbol{U}$ a matrix whose columns are $K-1$ orthonormal vectors. Using Lemma 15, we get that

$$\mu\sqrt{2} \leq \sigma_{\min}(\boldsymbol{H}_0) \leq \sigma_{\max}(\boldsymbol{H}_0),$$

where $\sigma_{\min}(\boldsymbol{H}_0), \sigma_{\max}(\boldsymbol{H}_0)$ denote its largest and smallest nonzero singular values. Then, since $\boldsymbol{z}_0 \in \text{conv}(\boldsymbol{H}_0)$ we have $\boldsymbol{z}_0 = \boldsymbol{H}_0^\top \alpha_0$ for some $\alpha_0$ s.t. $\alpha_0 \mathbf{1} = \mathbf{1}$. It holds,

$$\|\boldsymbol{z}_0\|_2 \leq \sigma_{\max}(\boldsymbol{H}_0)\|\alpha_0\|_2 \leq \sigma_{\max}(\boldsymbol{H}_0). \tag{8}$$

Note that

$$\sigma_{\max}(\widehat{\boldsymbol{H}} - \mathbf{1}\boldsymbol{z}_0^\top) \leq \sigma_{\max}(\widehat{\boldsymbol{H}}) + \sigma_{\max}(\mathbf{1}\boldsymbol{z}_0^\top) = \sigma_{\max}(\widehat{\boldsymbol{H}}) + \sqrt{K}\|\boldsymbol{z}_0\|_2. \tag{9}$$

Therefore, using Lemma 17 we have

$$\mathcal{D} \leq c\left[K^{3/2}\Delta_{1/2}\kappa(\boldsymbol{P}_0(\widehat{\boldsymbol{H}})) + \frac{\sigma_{\max}(\widehat{\boldsymbol{H}})\Delta_{1/2}K^{1/2}}{\mu} + \frac{K\Delta_{1/2}\|\boldsymbol{z}_0\|_2}{\mu}\right] + c\sqrt{K}\|\mathbf{F}\|_F, \tag{10}$$

where $\Delta_{1/2}$ equals $\Delta_1$ for problem (4) and $\Delta_2$ for problem (6), and $\kappa(\boldsymbol{A})$ stands for the conditioning number of matrix $\boldsymbol{A}$. In addition, Lemma 18 implies that

$$\mathcal{L}(\boldsymbol{H}_0, \widehat{\boldsymbol{H}})^{1/2} \leq \frac{1}{\alpha}\max\left\{(1+\sqrt{2})\sqrt{K}, \sqrt{2}\kappa(\boldsymbol{H}_0)\right\}\mathcal{D}. \tag{11}$$

**Step 3: Combining and final bound**

By Lemma 19 it holds

$$\mathcal{D} \leq c\Big[\frac{K^{3/2}\mathcal{D}\Delta_{1/2}}{\alpha(\mu - 2\Delta_{1/2})\sqrt{2}} + \frac{K^2\sigma_{\max}(\boldsymbol{H}_0)\Delta_{1/2}}{(\mu - 2\Delta_{1/2})\sqrt{2}} + \frac{\mathcal{D}K^{1/2}\Delta_{1/2}}{\alpha\mu}$$
$$+ \frac{\sigma_{\max}(\boldsymbol{H}_0)\Delta_{1/2}K}{\mu} + \frac{K\Delta_{1/2}\|\boldsymbol{z}_0\|_2}{\mu}\Big] + c\sqrt{K}\|\mathbf{F}\|_F. \tag{12}$$

We understand that $\mathcal{D} = \mathcal{O}_{\Delta_{1/2}\to 0}(\Delta_{1/2})$ and for small enough $\Delta_{1/2}$ there exists a constant $c > 0$ such that

$$\mathcal{D} \leq c\Delta_{1/2} + c\sqrt{K}\|\mathbf{F}\|_F$$

By (5) and (7), it yields that for small enough noise error $\|\mathbf{F}\|_F$ one has

$$\mathcal{D} \leq c\|\mathbf{F}\|_F,$$

for some (other) constant $c > 0$. Plugging this result in (11) we prove the result.

## B.3 Proof of Theorem 11

$\mathbf{N} \mapsto \nabla_{\mathbf{N}}h(\mathbf{H}, \mathbf{W}, \mathbf{N})$ is Lipschitz continuous with moduli $L = 2$. The statement follows from Proposition 4.1 in Javadi & Montanari (2020a) and from Theorem 1 in Bolte et al. (2014).

## C  Propositions and Lemmas

• Results that we can use directly from Javadi & Montanari (2020b): Lemma B.1, Lemma B.2, Lemma B.3.

• Results of Javadi & Montanari (2020b) that has to be adapted: Lemme B.4 (done in Lemma 15), Lemma B.5 (done in Lemma 16), and Lemma B.6 (done in Lemma 17).

**Proposition 14** *For* $\widehat{\boldsymbol{H}}$ *solution to* (4) *(or* (6)*) one has* $\widetilde{\mathcal{D}}(\widehat{\boldsymbol{H}}, \mathbf{X}) \leq \widetilde{\mathcal{D}}(\mathbf{H}_0, \mathbf{X})$.

**Proof.** Observe that $\overline{\mathcal{D}}(\mathbf{X}, \mathbf{H}_0) = \|\mathbf{F}\|_F^2$. By (5) and (7), $\mathbf{H}_0$ is feasible for (4) (or (6)) then $\widetilde{\mathcal{D}}(\widehat{\boldsymbol{H}}, \mathbf{X}) \leq \widetilde{\mathcal{D}}(\mathbf{H}_0, \mathbf{X})$

∎

**Lemma 15 (Adapted version of Lemma B.4 of Javadi & Montanari (2020b))** *If* $\boldsymbol{H}$ *is feasible for problem* (4) *(or* (6)*) and has linearly independent rows, then we have*

$$\sigma_{\min}(\boldsymbol{H}) \geq \sqrt{2}(\mu - 2\Delta_{1/2}),\tag{13}$$

*where* $\Delta_{1/2}$ *equals* $\Delta_1$ *for problem* (4) *and* $\Delta_2$ *for problem* (6).

**Proof.** Consider the notation and the outline of proof Lemma B.4 in Javadi & Montanari (2020b). The adaptation is simple here. The trick is to only consider rows in the training set, $\mathbf{T}_{\mathrm{train}}(\mathbf{X}_0)$: the indice $i$ of proof of Lemma B.4 in Javadi & Montanari (2020b) correspond to the $n - N$ first rows in our case (the training set); and one should replace $\mathbf{X}_0$ by $\mathbf{T}_{\mathrm{train}}(\mathbf{X}_0)$. This proof requires only feasibility of $\boldsymbol{H}$ and works no matter if a nonnegative constraint on $\boldsymbol{H}$ is active (as in Program (6)). ∎

**Lemma 16 (Adapted version of Lemma B.5 of Javadi & Montanari (2020b))** *For* $\widehat{\boldsymbol{H}}$ *solution to* (4) *(or* (6)*), it holds*

$$\widetilde{\mathcal{D}}(\widehat{\boldsymbol{H}}, \mathbf{X}_0)^{1/2} \leq \widetilde{\mathcal{D}}(\mathbf{H}_0, \mathbf{X}_0)^{1/2} + c\sqrt{K}\|\mathbf{F}\|_F.$$

**Proof.** Consider the notation and the outline of proof Lemma B.5 in Javadi & Montanari (2020b). Note that Eq. (B.103) holds by Proposition 14. Form Eq. (B.104), the proof remains unchanged once one substitutes $\mathcal{D}$ by $\widetilde{\mathcal{D}}$. ∎

**Lemma 17 (Adapted version of Lemma B.6 of Javadi & Montanari (2020b))** *For* $\widehat{\boldsymbol{H}}$ *the optimal solution of problem* (4) *(or* (6)*), we have*

$$\alpha(\mathcal{D}(\widehat{\boldsymbol{H}}, \boldsymbol{H}_0)^{1/2} + \mathcal{D}(\boldsymbol{H}_0, \widehat{\boldsymbol{H}})^{1/2}) \leq c\left[K^{3/2}\Delta_{1/2}\kappa(\boldsymbol{P}_0(\widehat{\boldsymbol{H}})) + \frac{\Delta_{1/2}\sqrt{K}}{\mu}\sigma_{\max}(\widehat{\boldsymbol{H}} - \mathbf{1}z_0^{\mathsf{T}})\right] + c\sqrt{K}\|\mathbf{F}\|_F\tag{14}$$

*where* $\boldsymbol{P}_0 : \mathbb{R}^d \to \mathbb{R}^d$ *is the orthogonal projector onto* $\mathrm{aff}(\boldsymbol{H}_0)$ *(in particular,* $\boldsymbol{P}_0$ *is an affine map), and* $\Delta_{1/2}$ *equals* $\Delta_1$ *for problem* (4) *and* $\Delta_2$ *for problem* (6).

**Proof.** Invoke the proof of Lemma B.6 in Javadi & Montanari (2020b) using the fact that $\widetilde{\mathcal{D}}(\mathbf{H}, \mathbf{X}) \leq \mathcal{D}(\mathbf{H}, \mathbf{X})$ and $\overline{\mathcal{D}}(\mathbf{X}, \mathbf{H}) \leq \mathcal{D}(\mathbf{X}, \mathbf{H})$. ∎

**Lemma 18** *Let* $\boldsymbol{H}, \boldsymbol{H}_0$ *be matrices with linearly independent rows. We have*

$$\mathcal{L}(\boldsymbol{H}_0, \boldsymbol{H})^{1/2} \leq \sqrt{2}\kappa(\boldsymbol{H}_0)\mathcal{D}(\boldsymbol{H}_0, \boldsymbol{H})^{1/2} + (1 + \sqrt{2})\sqrt{K}\mathcal{D}(\boldsymbol{H}, \boldsymbol{H}_0)^{1/2},\tag{15}$$

*where* $\kappa(\boldsymbol{A})$ *stands for the conditioning number of matrix* $\boldsymbol{A}$.

**Proof.** See Lemma B.2 in Javadi & Montanari (2020b) ∎

**Lemma 19** *It holds*

$$\kappa(\boldsymbol{P}_0(\widehat{\boldsymbol{H}})) \leq \left[\frac{\mathcal{D}}{\alpha(\mu - 2\Delta_{1/2})\sqrt{2}} + \frac{K^{1/2}\sigma_{\max}(\boldsymbol{H}_0)}{(\mu - 2\Delta_{1/2})\sqrt{2}}\right].$$

**Proof.** The proof is given by Equations B.189-194 in Javadi & Montanari (2020b). ∎

## D   Algorithms for mNMF

In this section we report *Block Coordinate Descend* (BCD) Algorithm (see Algorithm 3) and accelerated *Hierarchical Alternate Least Square* (HALS) for mNMF (see Algorithm 5), which is a generalization of Algorithm described in Gillis & Glineur (2012) to the matrix factorization with mask.

---

**Algorithm 3** BCD for mNMF

---

1: **Initialization**: choose $\mathbf{H}^0 \geq \mathbf{0}, \mathbf{W}^0 \geq \mathbf{0}$, and $\mathbf{N}^0 \geq \mathbf{0}$, set $i := 0$.
2: **while** stopping criterion is not satisfied **do**
3:     $\mathbf{H}^{i+1} := \text{update}(\mathbf{H}^i, \mathbf{W}^i, \mathbf{N}^i)$
4:     $\mathbf{W}^{i+1} := \text{update}(\mathbf{H}^{i+1}, \mathbf{W}^i, \mathbf{N}^i)$
5:     $\mathbf{N}^{i+1} := \text{update}(\mathbf{H}^{i+1}, \mathbf{W}^{i+1}, \mathbf{N}^i)$
6:     $i := i + 1$
7: **end while**

---

---

**Algorithm 4** ALS for mNMF

---

1: **Initialization**: choose $\mathbf{H}^0 \geq \mathbf{0}, \mathbf{W}^0 \geq 0$, set $\mathbf{N}^0 := \mathcal{P}_{\mathbf{X}}(\mathbf{H}^0 \mathbf{W}^0)$ and $i := 0$.
2: **while** stopping criterion is not satisfied **do**
3:     $\mathbf{H}^{i+1} := \min_{\mathbf{H} \geq 0} \|\mathbf{N}^i - \mathbf{W}^i \mathbf{H}\|_F^2$
4:     $\mathbf{W}^{i+1} := \min_{\mathbf{W} \geq 0, \mathbf{W}\mathbf{1}=\mathbf{1}} \|\mathbf{N}^i - \mathbf{W}\mathbf{H}^{i+1}\|_F^2$
5:     set $\mathbf{N}^{i+1} := \mathcal{P}_{\mathbf{X}}(\mathbf{W}^{i+1}\mathbf{H}^{i+1})$
6:     $i := i + 1$
7: **end while**

---

---

**Algorithm 5** accelerated HALS for mNMF

---

1: **Initialization**:  choose $\mathbf{H}^0 \geq \mathbf{0}, \mathbf{W}^0 \geq \mathbf{0}$, *nonnegative rank $K$*, and $\alpha > 0$. Set $\mathbf{N}^0 = \mathcal{P}_{\mathbf{X}}(\mathbf{W}^0 \mathbf{H}^0)$,
    $\rho_{\mathbf{W}} := 1 + n(m + K)/(m(K + 1))$, $\rho_{\mathbf{H}} := 1 + m(n + K)/(n(K + 1))$, and $i := 0$.
2: **while** stopping criterion is not satisfied **do**
3:     $\mathbf{A} := \mathbf{N}\mathbf{H}^{i\top}$, $\mathbf{B} := \mathbf{H}^i \mathbf{H}^{i\top}$
4:     **for** $k \leq k_{\mathbf{W}} := \lfloor 1 + \alpha \rho_{\mathbf{W}} \rfloor$ **do**
5:         **for** $\ell \in [K]$ **do**
6:             $C_\ell := \sum_{j=1}^{\ell-1} W_j^{k+1} B_{j\ell} + \sum_{j=\ell+1}^{K} W_j^k B_{j\ell}$
7:             $W_\ell^k := \max(0, (A_\ell - C_\ell)/B_{\ell\ell})$
8:         **end for**
9:         $\mathbf{W}^{k_{\mathbf{W}}} := \mathcal{P}_\Delta(\mathbf{W}^{k_{\mathbf{W}}})$
10:    **end for**
11:    $\mathbf{N} := \mathcal{P}_{\mathbf{X}}(\mathbf{W}^{k_{\mathbf{W}}} \mathbf{H}^i)$
12:    $\mathbf{A} := \mathbf{W}^{k_{\mathbf{W}}} \mathbf{N}$, $\mathbf{B} := \mathbf{W}^{k_{\mathbf{W}}\top} \mathbf{W}^{k_{\mathbf{W}}}$
13:    **for** $k \leq k_{\mathbf{H}} := \lfloor 1 + \alpha \rho_{\mathbf{H}} \rfloor$ **do**
14:        **for** $\ell \in [K]$ **do**
15:            $C_\ell := \sum_{j=1}^{\ell-1} H_j^{k+1} B_{j\ell} + \sum_{j=\ell+1}^{n} H_j^k B_{j\ell}$
16:            $H_\ell^k := \max(0, (A_\ell - C_\ell)/B_{\ell\ell})$
17:        **end for**
18:    **end for**
19:    $\mathbf{W}^{i+1} := \mathbf{W}^{k_{\mathbf{W}}}$, $\mathbf{H}^{i+1} := \mathbf{H}^{k_{\mathbf{H}}}$
20:    $\mathbf{N} := \mathcal{P}_{\mathbf{X}}(\mathbf{W}^{i+1}\mathbf{H}^{i+1})$
21:    $i := i + 1$
22: **end while**

---

# E  KKT conditions for mNMF

In this section we determine the KKT condition for mNMF problem, namely

$$\min_{\substack{\mathbf{W1}=\mathbf{1},\mathbf{W}\geq\mathbf{0} \\ \mathbf{H}\geq\mathbf{0} \\ \mathbf{T(N)=X}}} \|\mathbf{N} - \mathbf{WH}\|_F^2 =: \mathcal{F}(\mathbf{N}, \mathbf{W}, \mathbf{H})\,. \tag{mNMF}$$

Let us introduce the dual variables $\mathbf{V} \geq \mathbf{0}$, $\mathcal{G} \geq \mathbf{0}$, $t \in \mathbb{R}^n$, and $\mathbf{Z} \in \text{range}(\mathbf{\Pi})$ such that $\mathbf{T(Z) = Z}$. The Lagrangian of mNMF problem is

$$\mathcal{L}(\mathbf{N}, \mathbf{W}, \mathbf{H}, \mathbf{V}, \mathcal{G}, t, \mathbf{Z}) = \mathcal{F}(\mathbf{N}, \mathbf{W}, \mathbf{H}) - \langle \mathbf{W}, \mathbf{V} \rangle + \langle \mathbf{W1}_K - \mathbf{1}_N, t \rangle - \langle \mathbf{H}, \mathcal{G} \rangle - \langle \mathbf{N} - \mathbf{X}, \mathbf{Z} \rangle\,.$$

The KKT condition are the following:

$$\nabla_{\mathbf{N}}\mathcal{L} = \mathbf{N} - \mathbf{WH} - \mathbf{Z} = \mathbf{0} \qquad \Longleftrightarrow \mathbf{T(N - WH)} = \mathbf{Z} \wedge \mathbf{T^{\perp}(N - WH)} = \mathbf{0} \tag{16}$$

$$\nabla_{\mathbf{W}}\mathcal{L} = (\mathbf{WH} - \mathbf{N})\mathbf{H}^{\top} - \mathbf{V} - t\mathbf{1}_K^{\top} = \mathbf{0} \qquad \Longleftrightarrow \mathbf{V} = (\mathbf{WH} - \mathbf{N})\mathbf{H}^{\top} - t\mathbf{1}_K^{\top} \tag{17}$$

$$\nabla_{\mathbf{H}}\mathcal{L} = \mathbf{W}^{\top}(\mathbf{WH} - \mathbf{N}) - \mathcal{G} = \mathbf{0} \qquad \Longleftrightarrow \mathcal{G} = \mathbf{W}^{\top}(\mathbf{WH} - \mathbf{N}) \tag{18}$$

$$\langle \mathbf{W}, \mathbf{V} \rangle = \mathbf{0} \qquad \Longleftrightarrow \langle \mathbf{W}, \nabla_{\mathbf{W}}\mathcal{F} - t\mathbf{1}_K^{\top} \rangle = \mathbf{0} \tag{19}$$

$$\langle \mathbf{H}, \mathcal{G} \rangle = \mathbf{0} \qquad \Longleftrightarrow \langle \mathbf{H}, \nabla_{\mathbf{H}}\mathcal{F} \rangle = \mathbf{0} \tag{20}$$

From the complementarity conditions (19), it follows:

$$W_{i,j} > 0 \Longrightarrow V_{i,j} = 0 \Longrightarrow t_i = -(\nabla_W \mathcal{F})_{ij} \ \forall j$$

In order to compute $t_i$, we can select a row $W^{(i)}$, find any entry $W_{i,j} > 0$ and apply the previous formula. In the practical implementation phase, in order to make numerically more stable the estimation of $t_i$'s, we can adopt a slightly different strategy by averaging the values of $t_i$ computed per row entry $W_{i,j} > 0$.

