# OpenReview forum: "Time series recovery from partial observations via Nonnegative Matrix Factorization"
_TMLR — Rejected by TMLR_

### Review · Reviewer_rvWn · 2024-06-20

**Summary Of Contributions:**

The paper considers the time series prediction problem and formulates it as a nonnegative matrix factorization problem. The paper shows a sufficient condition that the factorization is unique and shows an analysis on the error. The proposed algorithms are based on previous work but with some twists. The paper shows experimental results using artificial and real time-series data sets.

**Audience:**

Yes

**Claims And Evidence:**

No

**Requested Changes:**

-Comparison with previous work

The paper does not discuss the difference between the present work and previous related work. Please clarify the differences so that the technical/original contribution becomes clear.


-On the formulation

In the formulation of the optimization problem, future entries are set to be zero. But, the objective measures the difference between predictions and zeros at future entries. It that a right formulation? It enforces the predictions towards zeros. I am afraid that future entries are not included in the objective. Please clarify this issue.

-On the generalization performance

Assumptions A1(A1’) and A2 ensure the uniqueness of the decomposition (Theorem 7) . I feel that the assumption is rather strong since it ensures uniqueness. Can you compare these assumptions with other assumptions used in previous related work?

What is the intuition of T_alpha-uniqueness? Without any explanation, it seems artificial and I wonder if Assumption A3 is natural.

Further, Theorem 7 and 10 do not take into account of the error on future instances since the given future entries are zero. This makes the theoretical guarantees weak. Please clarify this.

-Algorithm

The proposed algorithms and their convergence proofs are based on previous work. I guess the original part of the algorithm is to employ the linear operator \Pai. Please clarify what are original components.

-Experiments

The experiment section does not describe the details of the setting of the proposed and previous algorithms. For example, how to determine the size of the decomposition (say, K) is not shown, which would affect the results significantly. Please clarify the details so that one can reconstruct the experimental results.

-Notation

What does \ell \leq [K] mean? (In Theorem 10)

**Strengths And Weaknesses:**

Strengths
-The proposed algorithms have convergence guarantees.
-The decomposition is guaranteed to be unique under some technical conditions.
-The experimental results show that the proposed methods perform better than other alternatives.

Weaknesses:
-The analyses do not seem to guarantee generalization error on future instances.
-The proposed algorithms are based on previous work and the originality is unclear.
-There is no section discussing the difference between the previous and present work, which makes it difficult to evaluate its technical contribution.
-The experiment section lacks sufficient details to reconstruct the results. It makes the results unreliable.

---

> ### Author Response · Authors · 2024-07-09
> **Answer to Reviewer rvWn (1/2)**
>
> We thank the referee for her/his detailed comments and suggestions. We reviewer points several weaknesses:
>
>     •  The analyses do not seem to guarantee generalization error on future instances.
>
>
>     •  The proposed algorithms are based on previous work and the originality is unclear.
>
>
>     •   There is no section discussing the difference between the previous and present work, which makes it difficult to evaluate its technical contribution.
>
>
>     •  The experiment section lacks sufficient details to reconstruct the results. It makes the results unreliable. *
>
> on which we would like to answer:
>
> - We prove in Theorem 10 that some risk on the archetypes is smaller than the noise, then for each true archetype there exists an estimated archetype that is close to it. For sufficiently small noise, this means that there is a one-to-one correspondence between estimated archetypes and true archetypes, with small errors. Hence, our result shows that we recover the archetypal decomposition basis with an error of the same level as the noise. We will add a remark in the revised version to emphasize this point. Guarantee generalization error on future instance can be derived from this result. By triangular inequality, it is then possible to show that the generalization error of the order of the noise $||F||_F$. We focused on the error on the archetypes in the submission. The revised version will add a detailed comment on the generalization error.
> - The theoretical analysis and algorithms for "masked" AMF (mAMF) and "masked" and "normalized" NMF (mNMF) is new. We prove that the archetypes error estimation is of the order of the noise and the PALM algorithm converge to stationary points for these algorithms. We show that this method compares favorably with SOTA methods in time series forecasting. We agree that this main message is somehow hidden and we will fix the presentation in the revised version and add a section on related works to give a detailed comparison with the literature.
> - We agree with the referee comment on the experiment section. We will add sections presenting the datasets, the hyper-parameter tuning, the computational complexity and more experiments involving transformers.
>
> We agree with all the requested changes :
>
>
>     •  The paper does not discuss the difference between the present work and previous related work. Please clarify the differences so that the technical/original contribution becomes clear
>
> As mentioned above (and in our response to others referees), in the revised version, we will add a section on related works to give a detailed comparison with the literature.
>
>
>
>     •  In the formulation of the optimization problem, future entries are set to be zero. But, the objective measures the difference between predictions and zeros at future entries. It that a right formulation? It enforces the predictions towards zeros. I am afraid that future entries are not included in the objective. Please clarify this issue.
>
> The forecast values are set to zero arbitrarily, they are not used by the algorithm and does not affect the forecasted values by no means. We will add a comment to clarify this point in the revised version. We show that the archetypal reconstruction error (and generalization error in the revised version) is of the order of the noise error term.
>
>
>     •  Assumptions A1(A1’) and A2 ensure the uniqueness of the decomposition (Theorem 7) . I feel that the assumption is rather strong since it ensures uniqueness. Can you compare these assumptions with other assumptions used in previous related work? What is the intuition of T_alpha-uniqueness? Without any explanation, it seems artificial and I wonder if Assumption A3 is natural. Further, Theorem 7 and 10 do not take into account of the error on future instances since the given future entries are zero. This makes the theoretical guarantees weak. Please clarify this.
>
> These assumptions are standard in the literature, originating from the work of Donoho and Stodden "When Does Non-Negative Matrix Factorization Give a Correct Decomposition into Parts?" to the best of our knowledge. We will add more details on these assumptions, mentioning in particular "Theorems on positive data: On the uniqueness of NMF" by H Laurberg et al. for a closely related assumptions. In the submission we refer to "the notion of $\alpha$-uniqueness, introduced by Javadi & Montanari (2020a)" and we will give more emphasis on this point in the revised version commenting how the "mask" made us change the assumption.
>
>
>
>     •  Further, Theorem 7 and 10 do not take into account of the error on future instances since the given future entries are zero. This makes the theoretical guarantees weak. Please clarify this.
>
>
> - We answered to this point above. We apologize for the generalization error bound which will be derived a as corollary of the archetype estimation error bound (Theorem 10) in the revised version.

---

> ### Author Response · Authors · 2024-07-09
> **Answer to Reviewer rvWn (2/2)**
>
> (continued)
>
>     •  The proposed algorithms and their convergence proofs are based on previous work. I guess the original part of the algorithm is to employ the linear operator \Pai. Please clarify what are original components.
>
> The referee is right and we will clarify this point in the revised version.
>
>     •  "The experiment section does not describe the details of the setting of the proposed and previous algorithms. For example, how to determine the size of the decomposition (say, K) is not shown, which would affect the results significantly. Please clarify the details so that one can reconstruct the experimental results."
>
> We perform cross-validation to determine the size of the decomposition. In the revised version we will improve these comments by describing more in detail the hyper-parameters tuning procedure.
>
> The $[K]$ notation is a typo, it should be $\ell\leq K$ or equivalently $\ell\in[K]$.

---

### Review · Reviewer_tiXb · 2024-06-23

**Summary Of Contributions:**

The paper introduces the Sliding Mask Method (SMM), a novel algorithm for forecasting time series with possible missing entries. It leverages Nonnegative Matrix Factorization (NMF) and matrix completion to address these issues. The authors provide statistical guarantees on the uniqueness and robustness of solutions and validate the methodology through experiments on real-world and synthetic datasets.

**Audience:**

Yes

**Claims And Evidence:**

Yes

**Requested Changes:**

1. Include a comprehensive section on implementation details, including pseudocode and explanations of key steps, to facilitate reproducibility.
2. Add a thorough analysis of the computational complexity of the proposed method to provide insights into its scalability.
3. Include comparisons with more recent and advanced time series forecasting methods, such as transformer-based models, to highlight the method's advantages and limitations.
4. Provide detailed guidelines on the hyperparameter tuning process.
5. Revise the theoretical sections to improve clarity, possibly by adding more explanatory text and diagrams to illustrate complex concepts.

**Strengths And Weaknesses:**

Strengths:
1. The introduction of the Sliding Mask Method (SMM) is new, combining NMF with matrix completion to handle partially observed data efficiently.
2. The paper provides strong theoretical foundations, including proofs of uniqueness and robustness of the proposed solutions under partial observations. I do not verify the correctness of these specific proofs.
3. Empirical Validation: Extensive numerical experiments on real-world and synthetic datasets demonstrate the forecasting accuracy and efficiency of the proposed method.

Weaknesses:
1. The paper lacks detailed implementation instructions, making it challenging for others to reproduce the results independently.
2. There is limited discussion on the computational complexity of the proposed method, which is crucial for understanding its scalability.
3. While the paper includes comparisons with several baselines, it does not thoroughly discuss how the proposed method improves over the latest advanced methods, like transformer-based models for time series forecasting.
4. The process of selecting hyperparameters, particularly for the tuning parameter \lambda, is not adequately explained, which could affect the reproducibility of the results.
5. Some sections of the paper, particularly the theoretical proofs, are densely written and could benefit from clearer explanations and additional diagrams to aid understanding.

---

> ### Author Response · Authors · 2024-07-09
> **Answer to Reviewer tiXb**
>
> We thank the referee for her/his detailed comments and suggestions. We agree with all the requested changes which will be done in the revised version. About computational side, the reviewer underlines among the weaknesses of the paper that:
>
>     •  "The paper lacks detailed implementation instructions, making it challenging for others to reproduce the results independently."
>
> → As we stated in the overall response, we agree with the reviewer and we plan to include all the requested changes in a new version of this paper. We certainly include in Section 3 a detailed description of implementation instructions and in Appendix a description of the dataset characteristics, included its size, i.e., number of time series and time steps.
>
>     •  "There is limited discussion on the computational complexity of the proposed method, which is crucial for understanding its scalability."
>
> → It is true that we did not address the computational complexity of the proposed method. As mentioned in our response to Reviewer HT4J, we will add a section with the computational complexity. More details can be found in our response to Reviewer HT4J.
>
>
>     •  "While the paper includes comparisons with several baselines, it does not thoroughly discuss how the proposed method improves over the latest advanced methods, like transformer-based models for time series forecasting".
>
> → In the revised version, we will add a comparison with BasisFormer ion electricity dataset (we are finalizing the experiments).
>
>     •  "The process of selecting hyperparameters, particularly for the tuning parameter $\lambda$, is not adequately explained, which could affect the reproducibility of the results".
>
> → Regarding hyper-parameters tuning, we report in the paper that "Note that, both parameters $K$ and $\lambda$ can be tuned by cross-validation Arlot \& Celisse (2010), as done in our experiments, see Section 3". We will detailed hyper-parameters tuning in the revised version.

---

### Review · Reviewer_HT4J · 2024-06-29

**Summary Of Contributions:**

The work introduces a Sliding Mask Method, a nonnegative matrix factorization (NMF) based method aimed to forcesing time series with potentially partial observations. There are some theoretical guarantees on the uniqueness and robustness of solutions under partial observations (though I was not able to follow the arguments due to writing) and applications to real-world datasets. The algorithm seems to be fast and trainable on laptop computers, as is expected for NMF problems.

**Audience:**

Yes

**Broader Impact Concerns:**

No concerns.

**Claims And Evidence:**

No

**Requested Changes:**

I am unfortunately leaning to reject this work, which seems to be intended for a much longer journal format. After several reads, even though the main text consists of 10 pages, I believe the results and the assumptions should have been motivated more thoroughly, with dedicated discussions of their importance and/or relevance. In my humble opinion, the fact that almost all experimental results are pushed to the Appendix should have been a red flag for the authors, who could have opted for a longer, yet well presented submission. Please find below my major concerns and questions:

- I believe that there are several important work from literature that are not cited. For example, a simple google search revealed this relevant paper (http://proceedings.mlr.press/v70/mei17a/mei17a.pdf) which was not cited or discussed. The current work seems to be quite different in comparison, but a proper literature review on NMF and its applications is needed as a standalone section. There are also several work from dynamical system theory literature about recovery of temporal information from partial observations that may be fitting to discuss here.

- Another very important work, also not cited, is the following: https://www.nature.com/articles/s41567-023-02303-0. This work is very relevant for the authors' claims. For example, if I were to argue that the representations in the real-world may be high-dimensional, so why would the authors believe a low-rank reconstruction would work well, what would the response be? In its current form, no such response is present in the introduction of this paper.

- This brings me to the next point. There is no discussion section in this work. I believe the authors should do a much better job of connecting their results to the broader literature and discuss the implications.

- Most of the choices in Section 1.2 seem arbitrary and not motivated. For example, why are we interested in AMF? What makes it a good method? Why would we focus on NNMF vs NMF? Would the former be general enough to cover all possible solutions achieved by the latter by trivial transformations? If yes, state and prove. If well known, cite. If no, state for the reader and explain the motivation behind NNMF over NMF. K should also be explicitly defined as the rank of the matrix, which is alluded to in mathematical notation but not defined in words.

- The notation, when it was defined properly, was still quite confusing. For example, M is not noisy version of M^*, but of M_T^*? The block diagonal form at the start of page 3 does not define where the blocks are cut, though the reader is left to infer from the shape of B. But then, why is the linear map defined this way? The reader gets to learn at the end of the page. In general, motivating first and then stating the mathematical equations is probably a better approach to ease the readers into the notation and the story.

- The biggest question for me was the start of the page 6, where the authors causally drop a strong *periodicity* assumption for the time series. I do not understand how this is motivated, where this is used during training, or what happens if this is midly vs strongly violated. Can you please expand?

- For Section 2.2, could you explain how zeroing the future values for the training is reasonable? Would the predictions be the same if these values were set to 10, or 100? If not, how and why did the authors choose 0?

- For Section 2.3, why did the authors not use the ADMM method (https://web.stanford.edu/~boyd/papers/pdf/admm_distr_stats.pdf)? It seems to me naively that an ADMM based (2nd order, though the Hessian would likely be the correlation matrix?) solver might have been faster than PGD? If the new algorithms (Algo 1 and 2) are part of the contributions, it would be interesting to see the training speed compared to some basic baselines.

- Regarding Sections 3-4, as noted earlier, I was not able to follow the results here as they were not comprehensively represented or discussed.

Overall, I am convinced that there are new and interesting aspects of this work. However, the presentation made it extremely difficult for me, even after several reads, to understand the main results and put them in context with the known literature on low-rank hypothesis and NMF. I believe the work may eventually be accepted at TMLR, but that would require substantial revisions and perhaps a new submission that is more representative (in length) of the amount of work involved (e.g., for the reviewers).

**Strengths And Weaknesses:**

## Strengths

Given the computational efficiency of non-negative matrix factorization and the recent theoretical and empirical work (though not cited here) supporting the abundance of low-rank representations in the real-world, a new attempt at forecasting problems with NMF is both timely and interesting. The work satisfies the criteria for being of interest to the TMLR audience.

## Weaknesses

The biggest weakness, for me, was the presentation. I was not able to follow most of the arguments due to how dense, and unfortunately sometimes incoherent, writings. Most choices seemed unmotivated and arbitrary; notations undefined, and important details (e.g. experimental results) stuffed to appendix. As it stands, the work is not clearly presented and the claims cannot be evaluated without substantial revisions.

---

> ### Author Response · Authors · 2024-07-09
> **Response to Reviewer HT4J (1/2)**
>
> We thank the referee for her/his detailed comments and suggestions. We agree with enhancing the paper with more comments, clarifications and readability.
>
>      •  For example, if I were to argue that the representations in the real-world may be high-dimensional, so why would the authors believe a low-rank reconstruction would work well, what would the response be? In its current form, no such response is present in the introduction of this paper.
>
> → We thank the reviewer for pointing to this aspect of real data representation. Sparse or Low-Rank representations are ubiquitous in applications and well studied in the literature. In this submission, we collect time series observations over a period of time with fixed length. Hence a time series is cut into several smaller time series with same length. For instance, observing sales over a period of one year, one can consider 52 weekly time series (one per week). These observations are the rows of our observed matrix. The nonnegative low rank hypothesis assumes that the weekly time series of the dataset can be decomposed as a sum of $K$ archetypal time series plus an error term. Of course, this error term can model the model bias. The $K$ archetypal time series are learnt on the entire data set, leading to a decoding basis learnt from the entire dataset. This technique can be seen as dimension reduction, each observation can be summarized as $K$ weights such that the resulting weighted sum of archetypes is a good approximation of the observation. This point of view can be found in the literature as in Javadi \& Montanari (2020a) for instance. The relevance of such hypothesis on real data cannot be proven beforehand. Our numerical study on real data shows that we achieve good results in prediction, better than standard methods in time series analysis (SARIMAX, EXP, RFR, LSTM). It suggests that the low rank assumption is reasonable for the datasets studied in the paper. In the revised version, we will extend our comparison to recent advances in transformers for time series analysis. The experiments are presently being conducted and we will report them in the revised version. We will include a section discussing the low rank hypothesis in the revised version.
>
>
>      •  Most of the choices in Section 1.2 seem arbitrary and not motivated. For example, why are we interested in AMF? What makes it a good method? Why would we focus on NNMF vs NMF? Would the former be general enough to cover all possible solutions achieved by the latter by trivial transformations?
>
> → AMF is well studied in Javadi \& Montanari (2020a) and we referred to this paper for further details. This submission gives several strategies for matrix factorization given in Table 7. The "normalized" and "masked" factorization are new and introduced in the submission. The main contribution is to show that the estimation error on the archetypes is of the order of the noise term (Theorem 10) in the small noise regime. Hence, the masked AMF and masked NMF uncover a good approximation of the true archetypes. The archetypal analysis can be seen as a low rank representation which can be suited to forecasting real data time series as explained above. The normalized method (NNMF) interprets the weights of each observed time series as a point on the simplex (due to normalization) which can be interpreted as weights of a convex combination. Hence, each estimation of NNMF is a convex combination of the archetypes. This is not the case in NMF. Hence, this extra structure of NNMF makes it impossible to cover all possible solutions achieved by the former by trivial transformations of the latter and conversely.
>
>
>     •  "I believe that there are several important work from literature that are not cited. "
>
> → The referee is right. A standalone section with a more in-depth review will be given in the revision. The authors were aware of the paper cited by the reviewer. We decided to keep the related works section light, may be too light. In the revision we will include these works and more references to the literature on NMF citing recent advances and key books on the subject.
>
>
>      •  This brings me to the next point. There is no discussion section in this work. I believe the authors should do a much better job of connecting their results to the broader literature and discuss the implications.
>
> → A new section will be added in the revision giving a discussion connecting to broader literature. In particular, the links with NMF, archetypal analysis and transformers.
>
>     •  The notation, when it was defined properly, was still quite confusing.
>
> → We plan to present first the sliding method and its performances on real datasets. Then we will dig into the mathematical analysis and the theoretical results for the general framework adding clearer notations along the analysis to improve the readiness of the paper.

---

> ### Author Response · Authors · 2024-07-09
> **Response to Reviewer HT4J (2/2)**
>
> (continuing)
>
>
>     •  The biggest question for me was the start of the page 6, where the authors causally drop a strong periodicity assumption for the time series. I do not understand how this is motivated, where this is used during training, or what happens if this is mildly vs strongly violated. Can you please expand?
>
> → This is the beginning of the sliding mask method. It will be motivated earlier in the revised version, before the general framework. It will increase the readability of the paper. As for the referee question on model bias, if the periodicity is violated then one may expect to increase the number $K$ of archetypes (representations) to capture this non-periodic phenomenon. Highly non-periodic time series may cause a drop of performances as the archetypal modeling fails to capture it.
>
>
>     •  For Section 2.2, could you explain how zeroing the future values for the training is reasonable? Would the predictions be the same if these values were set to 10, or 100? If not, how and why did the authors choose 0?
>
> → The algorithm does not have access to these values. We put them to 0 arbitrarily.
>
>     •  For Section 2.3, why did the authors not use the ADMM method? It seems to me naively that an ADMM based (2nd order, though the Hessian would likely be the correlation matrix?) solver might have been faster than PGD? If the new algorithms (Algo 1 and 2) are part of the contributions, it would be interesting to see the training speed compared to some basic baselines.
>
> → Theorem 11 shows that PALM converges to critical points of the risk function. We will add a section of the computational complexity and running time of the algorithm. The costliest step is the projection onto the convex hull of $N$ (line 4). This projection can be computed by means of Frank-Wolfe algorithm in $O(N^2/\epsilon)$ to achieve a $\epsilon$-approximate solution. One can also use an active set method whose complexity is $O(AN/\epsilon)$ where $A$ is the dimension of the face on which the point is projected. The step 3 has complexity $O(NKT)$ (matrix multiplication). Projection on the simplex is $O(K\log K)$ so step 5 is $O(NK\log K)$. The last projection amounts to set to $X_{i,j}$ when the indices ${i,j}$ is observed, hence its cost is $O(NT)$. We initialize the PALM algorithm using a truncated spectral estimates which cost $O(NKT)$. The overall complexity on the dimension of one iteration of the PALM algorithm (and of its initialization) is $O(NK(T+\log K))$ when the projection lies on a face of dimension $A$ such that $A=O(K(T+\log K))$. The convergence rate of PALM is $O(1/t^2)$ after $t$ iterations (see Section 4 of "On the rate of convergence of the proximal alternating linearized minimization algorithm for convex problems" by Shefi et Teboulle). One gets a $\epsilon$-approximate stationary point in $O(NK(T+\log K)/ \sqrt{\varepsilon})$ operations.
>
>
> ADMM can be viewed as a dual variant of Douglas-Rachford’s splitting methods and hence an instance of proximal gradient algorithm. Although various variants of ADMM have been studied in the literature, their three main steps
> (two primal subproblems, and one dual update) remain the same in most existing papers. The ADMM algorithm was not studied for AMF or mNMF. In these cases, the two primal subproblems are quadratic programs with inequality constraints which can be time consuming to solve. Proximal variants of ADMM exist and their iterations would be very similar to the PALM iterations and would have the same computational complexity (this is not investigated by the paper). Moreover ADMM has a slower convergence rate of $O(1/t)$ than PALM. But this result should be understood as no better convergence results have been proven for our case (which is out of the scope of the submission). Hence, it is not clear that ADMM would perform better than PALM for our problem.
>
>     •  "Regarding Sections 3-4, as noted earlier, I was not able to follow the results here as they were not comprehensively represented or discussed."
>
> - We fundamentally agree with the reviewer that the computational experiments description is too short and this may cause a difficulty in following the results. As we state in the paper, we use both synthetic and real-world datasets to test our novel algorithms against state-of-the-art approaches and the computational results we obtain are quite promising (see Tables 1-4). In the future version of this paper, we aim to include a comprehensive section on implementation details and results discussion. We plan also to include a section with a clear description of the datasets used in computational experiments providing also the sizes of those datasets.

---

> > ### Comment · Reviewer_HT4J · 2024-07-09
> >
> > I thank the authors for the extensive rebuttal and detailed answers. To clarify, though my comments are posed as questions, my hope is that the authors will revise the manuscript to answer these (and similar) questions. A casual reader of TMLR should be able to understand the main points of the work and appreciate the contributions to the literature even if they are not following the immediately relevant literature or are not domain experts.
> >
> > Overall, the authors have the tendency in their responses (to my comments and others) to argue that certain things are standard in the literature and well known. This is not a substitute for a self-sufficient presentation, which is the main weakness I raised. I am looking forward to reading the revised manuscript, and am quite open to changing my mind if there is a *substantial* improvement in the clarity of the presented claims and evidence.

---

> > > ### Comment · Reviewer_HT4J · 2024-07-22
> > > **Final feedback**
> > >
> > > As I was writing my initial review, I had truly believed that this work would eventually be accepted. However I am unfortunately surprised by the poor revision performance by the authors.
> > >
> > > Unfortunately, the authors have not provided satisfactory revisions and skipped parts of real criticism in my review. It seems that the authors are unwilling to put actual work into their manuscript to put it into an easily accessible format for a general reader, instead they opted for additional experiments that were not necessary for acceptance or requested by all reviewers. Instead, all reviewers had issues on the clarity, which the authors had not properly addressed.
> > >
> > > As a low level feedback, I believe the authors should have made it clear how our comments led to changes in the manuscript, which would have allowed me to judge the outcome better even without the revised version posted. Moreover, it is not appropriate to leave the revision to the last five days, as reviewers should not be expected to read the revised manuscript in 1-2 days.

---

### Author Response · Authors · 2024-07-09
**Overall response**

First of all we would like to thank the reviewers for their comments and remarks, which will improve the presentation and the readability of the paper. We received two types of comments: one type addressing the way we present and discuss the results, and another type of remarks addressing the computational side of the paper, mainly related to the reproducible aspect of the numerical experiments and comparison to transformer methods.

For the first point, it appears from this round of discussion that the paper needs to be reshaped following two recommendations: first, more discussions on the hypotheses, the model, the results and the related works, the computational complexity and running time of the algorithm; second, more readability on notations and modeling, this can be achieved starting from the sliding mask method and then presenting the general framework (the submission was done the other way around).


For the second point, we fundamentally agree with the objections raised by the reviewers, and we plan to submit a revised version of the paper where we deal with those issues, by adding to Section 3 a section discussing in more detail the characteristics of the datasets, both for synthetic and real-world ones, and the algorithmic hyper-parameters tuning (in particular the rank $K$ of the output matrices). Moreover, we will add (soon) a comparison with recent advances in transformers for time series analysis using BasisFormer on electricity dataste as in SOTA results:

* @inproceedings{ni2023basisformer,
  title={{Basisformer}: Attention-based Time Series Forecasting with Learnable and Interpretable Basis},
  author={Ni, Zelin and Yu, Hang and Liu, Shizhan and Li, Jianguo and Lin, Weiyao},
  booktitle={Advances in Neural Information Processing Systems},
  year={2023}
}

We try to sort the questions of the referees starting from the most critical from our point of view in our detailed responses to each reviewer.

---

### Decision · Action_Editor_Ywgs · 2024-09-13

**Recommendation:** Reject

**Comment:**

Reviewers believe that a new attempt at forecasting problems with NMF is both timely and interesting. However, there are a few critical concerns. First of all, it is criticized that the paper is NOT easy to read since the presentation is dense and sometimes the writing is not coherent. Unfortunately, the author response was not satisfactory, so reviewers feel that the paper should go substantial revision. A much longer journal format is a better venue of this paper.
The paper lacks detailed implementation instructions, making it challenging for others to reproduce the results independently.
I hope authors found the review comments informative and can improve their paper by addressing these carefully in future submissions.

**Audience:**

As pointed out by a reviewer, "a casual reader of TMLR should be able to understand the main points of the work and appreciate the contributions to the literature even if they are not following the immediately relevant literature or are not domain experts". However, it is criticized that the presentation is dense and sometimes the writing is not coherent.

**Claims And Evidence:**

Nonnegative matrix factorization (NMF) has been extensively studied in the perspectives of algorithms, underlying theories, and applications. This paper presents a forecasting method based on NMF and its ability of matrix completion.  It provide statistical guarantees on the uniqueness and robustness of solutions and demonstrates its validity on synthetic and real-world datasets.